# Tsunami hazard and risk assessment for multiple buildings by considering spatial correlation of wave height using copulas

Yo Fukutani[1], Shuji Moriguchi[2], Kenjiro Terada[2], Takuma Kotani[3], Yu Otake[4], Toshikazu Kitano[5]

[1]College of Science and Engineering, Kanto Gakuin University, Yokohama, 236-8501, Japan
5   [2]International Research Institute of Disaster Science, Tohoku University, Sendai, 980-8572, Japan
[3]Nippon Koei Co., Ltd, Ibaraki, 300-1259, Japan
[4]Faculty of Engineering, Niigata University, Niigata, 950-2102, Japan
[5]Civil and Environmental Engineering, Nagoya Institute of Technology, Nagoya, 466-8555, Japan

10   *Correspondence to*: Yo Fukutani (fukutani@kanto-gakuin.ac.jp)

**Abstract.** It is necessary to evaluate aggregate damage probability to multiple buildings when performing probabilistic risk assessment for the buildings. The purpose of this study is to demonstrate a method of tsunami hazard and risk assessment for two buildings far away from each other, using copulas of tsunami hazards that consider the nonlinear spatial correlation of tsunami wave heights. First, we simulated the wave heights considering uncertainty by varying the slip amount and fault depths. 15   The frequency distributions of the wave heights were evaluated via the response surface method. Based on the distributions and numerically simulated wave heights, we estimated the optimal copula via maximum likelihood estimation. Subsequently, we evaluated the joint distributions of the wave heights and the aggregate damage probabilities via the marginal distributions and the estimated copulas. As a result, the aggregate damage probability of the ninety-ninth percentile value was approximately 1.0 % higher and the maximum value was approximately 3.0 % higher while considering the wave height correlation. We 20   clearly showed the usefulness of copula modeling considering the wave height correlation in evaluating the probabilistic risk of multiple buildings. We only demonstrated the risk evaluation method for two buildings, but the effect of the wave height correlation on the results is expected to increase if more points are targeted.

## 1 Introduction

Probabilistic hazard and risk assessment methods of disasters are developed mainly in the field of nuclear safety focused on 25   countermeasures relative to severe accidents at nuclear power plants. Among them, a variety of probabilistic tsunami hazard assessment (PTHA) and probabilistic tsunami risk assessment (PTRA) methods for tsunami disasters were rapidly developed since the 2000s (e.g., Geist and Parsons, 2006; Annaka et al., 2007; González et al., 2009; Thio et al., 2010; Løvholt et al., 2012; Goda et al., 2014; Fukutani et al., 2015; Løvholt et al., 2015; Park and Cox, 2016; De Risi and Goda, 2017; Grezio et al., 2017; Davies et al., 2018). The main purpose of a PTHA is to assess the likelihood of a given measure of tsunami hazard 30   metrics (e.g. maximum tsunami wave height) being exceeded at a particular location within a given time period. The most basic outcome of such an analysis is typically expressed as a hazard curve, which shows the exceedance level of the hazard

metric with the probability. This is often expressed as a rate of exceedance per year. A PTHA can be expanded to a PTRA by combining hazard assessment with loss evaluation of a target. Several studies have proposed a method of PTRA for an individual site in a local area. Detailed risk assessment is undoubtedly important in terms of grasping the risk of exposing assets located in a local area.

However, probabilistic risk evaluation methods are also utilized in cases to evaluate risks for multiple buildings (e.g., Kleindorfer and Kunreuther, 1999; Chang et al., 2000; Grossi and Kunreuther, 2005; Goda and Hong, 2008; Salgado-Gálvez, 2014; Scheingraber and Käser, 2019). With respect to businesses that own a building portfolio, including factories and offices over a wide area, it is extremely important in risk-based management decisions to evaluate the detailed risks posed by the building portfolio. A portfolio means a collection of assets held by an institution or a private individual. By quantitatively
assessing the risks posed by the building portfolio, for example, it is possible to identify assets held that have a large impact on the overall risk, and to compare the amount of risk held over time, which leads to support for decision-makers.

When evaluating physical risks for multiple buildings over a wide area, it is necessary to evaluate the aggregate risk for the buildings that are located at a distance. In these types of cases, it is necessary to evaluate the risk by considering the spatial correlation of hazards. For example, let us consider assessing the risk of two buildings located at two sites. When the positive
correlation of hazards between two sites is strong, the hazard at one site tends to be large if the hazard at another site is large. In this case, the hazards at the two target sites both increase, and as a result, the aggregate risk for the two buildings considering the hazard correlation increases. Conversely, when the positive correlation of hazards is small, the hazard at one site is not necessarily large, even if the hazard at another site is large. In this case compared to the former case, the hazards at the two target sites are smaller, and as a result, the aggregate risk for the two buildings is smaller if we assume that the vulnerability
of the two buildings is equal. Therefore, analyses that do not consider the spatial correlation of hazards involves the risk of underestimating the risk over a wide area. It is clear that the difference of aggregate risk between two cases becomes more prominent as the number of target sites increases. Analyses that consider the spatial correlation of hazards are relatively advanced in the field of earthquake hazard and risk assessment (e.g., Boore et al., 2003; Wang and Takada, 2005; Park et al., 2007) albeit insufficient in the field of tsunami hazard and risk assessment. Analyses that consider the hazard correlation using
copulas are used in hydrological/earthquake modeling (e.g., Goda and Ren, 2010; Goda and Tesfamariam, 2015; Salvadori et al., 2016;) although there is a paucity of the same in tsunami modeling.

In this study, we assume the occurrence of a large earthquake in the Sagami Trough in Japan that significantly affects the metropolitan area and evaluate the tsunami risk of two buildings located at distant locations by considering the spatial correlation of the tsunami wave height between the two sites. The objective of this study involves evaluating the frequency
distribution of the tsunami height via the response surface method and evaluating the spatial correlation of the tsunami heights and damages by using various copulas. Specifically, we analyze the frequency distribution (marginal distribution) of tsunami height via the response surface method and target two steel buildings located at Oiso and Miura along the Sagami Bay, Kanagawa Prefecture in Japan. Subsequently, we derive the joint distribution of tsunami wave heights between two sites by using various copulas and the marginal distributions, convert it to the joint distribution of damage via applying a damage

function, and evaluate the expected value of the aggregate damage probability for the target buildings. Finally, we confirm the extent to which the expected value of the aggregate damage probability fluctuates in a case where the spatial correlation of tsunami wave height is considered and a case where it is not considered.

The second chapter provides an outline of the response surface method and tsunami hazard and risk assessment method for multiple buildings using copulas. The third chapter describes a case where the proposed method is applied to the Sagami Trough area. The final conclusions are discussed in the fourth chapter.

## 2 Methodology

Figure 1 shows a flowchart of tsunami hazard and risk assessment considering the correlation of tsunami wave heights in this study. Herein, the risk assessment target points only correspond to two points: Oiso and Miura, Kanagawa prefecture in Japan. Figure 2 shows the location of these points. First, we simulate the tsunami wave heights considering the uncertainty at the target sites by numerical tsunami simulations via nonlinear long wave equations. Based on this, we construct a response surface and apply probability distributions to obtain a frequency distribution of tsunami wave heights. This distribution becomes a marginal distribution for a joint distribution of tsunami wave heights of two target points. Separately, we estimate appropriate copula via maximum likelihood estimation from the simulation results of the tsunami wave height considering uncertainty. Subsequently, we obtain a joint distribution of tsunami wave heights from the estimated copula and the marginal distributions of tsunami wave height. Furthermore, we obtain a joint distribution of damage probabilities by applying the tsunami damage function.

The outline of the response surface method and copula modeling used in this study is explained below. The response surface method is a statistical combination method to determine an optimum solution using the least number of measurement data possible. The basic idea is based on a reliability-based design scheme developed in the research field of geomechanics (e.g. Honjo, 2011). Generally, the response surface model is given by Eq. (1) as follows:

$$y = f(x_1, x_2, \dots, x_n) + \varepsilon \tag{1}$$

where explanatory variables correspond to $x_i$ ($i$ = 1, 2, 3, ..., $n$), response (object variable) corresponds to $y$, and error corresponds to $\varepsilon$. It should be noted here that a response surface is generated for a certain point. Therefore, it is necessary to generate a large number of response surfaces with spatial meshes in order to evaluate the spatial inundation height and flow depth variability, but such analysis is outside the scope of this study. Tsunami hazard assessment has many uncertainties in each process of tsunami generation, propagation, and run-up. Even considering only the earthquake source parameters that are the basis for calculating the initial displaced water level of the tsunami, there are fault length, fault width, fault depth, slip amount, rake, strike, and dip. The temporal and spatial changes of all these parameters more or less affect the tsunami hazard assessment. Numerous studies on the effect of earthquake source parameters on the initial displaced water level of tsunamis have been conducted (e.g., Hwang and Divoky 1970; Ward 1982; Ng et al. 1991; Pelayo and Wiens 1992; Whitmore 1993; Geist and Yoshioka 1996; Geist 1999; Geist 2002; Song et al. 2005). These studies reported that fault slip was an important

factor governing tsunami intensity. In addition, the Sagami Trough, which is the target earthquake of this study, has a complex crustal structure in the area where the Pacific Plate, the Philippine Sea Plate, and the North American Plate meet. Therefore, the depth where the Sagami Trough earthquake occurs is considered uncertain. Therefore, in this study, we decided to consider only the tsunami hazard uncertainty caused by the changes of slip amount and fault depth as an example. The heterogeneity of fault slip is an equally important factor, but we did not consider non-uniform slip distribution for purposes of simplicity. It is an important issue in the future to evaluate the heterogeneity of fault slip by response surface methodology. This is true for both slip heterogeneity and other fault parameters. For the above reasons, we model maximum tsunami wave height considering tsunami wave uncertainty with Eq. (2) after conducting tsunami numerical simulation with a nonlinear long wave equation. This formula is following the tsunami hazard evaluation method proposed by Kotani et al. (2016) that applied a reliability analysis framework using the response surface method proposed in Honjo (2011). The expression is as follows:

$$h(S, D) = aS + bD + cSD + dS^2 + e \qquad (2)$$

where $h(S, D)$ denotes the tsunami wave height, $S$ denotes the slip, $D$ denotes the fault depth, and $a$, $b$, $c$, $d$, and $e$ denote the undetermined coefficients. It should be noted that an error term is not included in Eq. (2). An example of the error term is to consider an error due to modeling. For example, Kotani et al. (2016) quantified the modeling error as the difference between the observed tsunami height and the numerically simulated tsunami height. The modeling error of the numerical analysis was also considered as one of the tsunami hazard uncertainties. However, the main purpose of this study is to propose a tsunami damage assessment method for multiple buildings using copula considering wave height correlation. Therefore, the modeling error is also ignored for simplification in this study.

This response surface method has an advantage that the probability distribution of the objective variable can be easily evaluated by applying an appropriate probability distribution to the explanatory variable and performing Monte-Carlo simulation. Although tsunami numerical simulation considering uncertainty usually has high calculation cost to conduct vast numbers of simulation cases, it is possible to significantly reduce the simulation cost by using the response surface method.

The foundation of the copula theory corresponds to the Sklar theorem (Sklar, 1959). A copula is a multivariate distribution whose marginals are all uniform over [0, 1]. Given this in combination with the fact that any continuous random variable can be transformed to be uniform over [0, 1] by its probability integral transformation, copulas are used to separately provide multivariate dependence structure from the marginal distributions. Let $F$ be a n-dimensional distribution function with marginals $F_1, \ldots, F_n$ and $H$ be a joint distribution function. There exists a n-dimensional copula $C$ such that for all $x$ in the domain of $F$, the following expression holds (Sklar, 1959):

$$H(x_1, \ldots, x_n) = C\{F_1(x_1), \ldots, F_n(x_n)\} = C(u_1, \ldots, u_n) \qquad (3)$$

where $u_i = F_i(x_i) \in [0,1]$, $i = 1, \ldots, n$. Figure 3 shows a simple synthetic example of a copula in a bivariate case. Fig. 3 (a) is a joint distribution function, Figs. 3 (b) and (c) are distribution functions of each variable (marginal distributions) and Fig. 3 (c) is a copula distributed over [0, 1]. Joe (1997) and Nelsen (1999) proposed the two comprehensive treatments on the topic. The two most common elliptical copulas correspond to the Gaussian copula and the t-copula whose copula functions in the bivariate case correspond to Eqs. (4) and (5).

$$C(u_1, u_2) = \Phi_\Sigma(\Phi^{-1}(u_1), \Phi^{-1}(u_2)) \tag{4}$$

$$C(u_1, u_2) = t_{\Sigma,v}(t_v^{-1}(u_1), t_v^{-1}(u_2)) \tag{5}$$

The Gaussian copula is simply derived from a multivariate Gaussian distribution function $\Phi_\Sigma$ with mean zero and correlation matrix $\Sigma$ by transforming the marginals by the inverse of the standard normal distribution function $\Phi$. The t-copula is derived in the same way as the Gaussian copula. Given a multivariate centered t-distribution function $t_{\Sigma,v}$ with correlation matrix $\Sigma$, $v$ degrees of freedom and with marginal distribution function $t_v$. The Archimedean copula is a widely-used copula family. The Archimedean copulas include the Gumbel, Frank, and Clayton copulas whose copula functions in the bivariate case correspond to Eqs. (6), (7), and (8), respectively, as follows:

$$C_\theta(u_1, u_2) = exp\{-[(-lnu_1)^\theta + (-lnu_2)^\theta +]^{1/\theta}\}, \; \theta \geq 1 \tag{6}$$

$$C_\theta(u_1, u_2) = -\frac{1}{\theta} ln\left\{1 + \frac{(exp(-\theta u_1)-1)(exp(-\theta u_2)-1)}{exp(-\theta)-1}\right\}, \; -\infty < \theta < \infty \tag{7}$$

$$C_\theta(u_1, u_2) = (u_1^{-\theta} + u_2^{-\theta} - 1)^{-1/\theta}, \; \theta \geq 1 \tag{8}$$

The Gumbel and Clayton copulas capture upper tail dependence and lower tail dependence, respectively, while the Frank copula does not exhibit tail dependence. Specifically, $\theta$ is estimated based on the maximum log-likelihood method. The copulas denote the symmetrical property with respect to diagonal lines of a unit square. To handle asymmetrical data in transformed space, we used an asymmetrical extreme-value copula (Tawn, 1988; Genest and Favre 2007; Genest and Segers, 2009). Extreme-value copulas are characterized by the dependence function $A$ as given in Eq. (9):

$$C(u_1, u_2) = exp\left[log\,(u_1 u_2) A\left\{\frac{log\,(u_1)}{log\,(u_1 u_2)}\right\}\right] \tag{9}$$

An asymmetric model using the copula with three parameters as mentioned by Tawn (1988) is given by:

$$A(t) = \{\theta^r(1-t)^r + \varphi^r t^r\}^{1/r} + (\theta - \varphi)t + 1 - \theta \tag{10}$$

where, $r$, $\theta$ and $\varphi$ are estimated based on the maximum log-likelihood method. The special case $\theta = 1$ and $\varphi = 1$ corresponds to the symmetric model proposed by Gumbel (1960), and thus this is termed as the asymmetric Gumbel copula. We use this copula for modeling asymmetrical data dependence.

In this study, we use bivariate as the tsunami wave height at two target points and model the correlation using copula. The linear correlation coefficient (Pearson's correlation coefficient) is an index that captures the linear relation between variables and essentially cannot express the dependency between variables that are not in linear relation. Conversely, the copula is a function that expresses the correlation based on the order of the data of each variable rather than the data itself. The order of the data is expressed by Kendall's $\tau$ (Kendall, 1938). Therefore, it is possible to quantify the nonlinear correlation between the variables. Table 1 shows theoretical value of Kendall's $\tau$ corresponding to the bivariate copulas and their parameter vectors. In this study, we show a simple evaluation method for two target points, although correlation between more points can be considered by using copulas.

## 3 Application to the Sagami Trough area

In this chapter, we demonstrate a case study where the hazard and risk assessment method described in the previous chapter is applied for two buildings located on the coast of Sagami Bay, Kanagawa prefecture in Japan. Section 3.1 shows the assessment target points, Section 3.2 shows the tsunami numerical simulation considering uncertainties, Section 3.3 constructs the response surface, Section 3.4 shows the modeling of tsunami wave height correlation using copulas, and Section 3.5 shows the results of the evaluation and discussion.

### 3.1 Risk assessment targets

Figure 2 (a) shows major subduction-zone earthquakes around the Japanese islands, namely the Sagami Trough earthquake, the Nankai Trough earthquake and Tohoku-type earthquake announced by NIED (2017). Figure 2 (b) shows the located points of tsunami hazard and risk assessment targets, namely Oiso and Miura, Kanagawa prefecture in Japan. The Sagami Trough earthquake covers most of the Kanto region, including the target points. Oiso is located at the approximate center of Sagami Bay coast, and Miura is located at the tip of the Miura Peninsula, which is located between Tokyo Bay and Sagami Bay. We assume a steel-framed building located at these two points and evaluate tsunami damage probability for the two buildings.

### 3.2 Tsunami numerical simulation considering uncertainties

In this section, we evaluate the tsunami wave heights by considering the uncertainty at the target points.

We selected ten earthquake occurrence sources of the Moment magnitude (Mw) 8 class along the Sagami Trough, which significantly affect the metropolitan area in Japan. The Sagami Trough is a 300 km long boundary between the Philippine Sea and North American plates. The assumed earthquake sources are shown in Fig. 4 (a). There are 10 earthquake sources and the Mw of the source's ranges from Mw 7.9 to Mw 8.6. The source 8 has maximum Mw 8.6. The sources are used for probabilistic ground motion prediction in Japan published by NIED (2017), and thus they exhibit 0.7 % occurrence probability in the next 30 years, and the weights of occurrence probability for each earthquake source. Table 2 shows the number of small faults in each source. Each small fault corresponded to a 2.5 km square, and the slip amount of the fault was set to a uniform value based on the moment magnitude (Mw) of each earthquake by using the following scaling laws of earthquakes according to Kanamori (1977):

$$Mo = \mu SA \tag{11}$$

$$Mw = \frac{log_{10}Mo - 9.1}{1.5} \tag{12}$$

where $Mo$ denotes moment magnitude (Nm), $\mu$ denotes shear modulus (Pa), $S$ denotes slip amount (m) and $A$ denotes earthquake source area (m$^2$). $\mu$ was set to $3.4 \times 10^{10}$ (Pa). In this study, we did not consider non-uniform slip distribution for purposes of simplicity. We set other fault parameters (i.e., fault depth, dip, rake, and strike) to the sources based on information published by the Cabinet Office (2013) in Japan, which were created from the crustal structure of data of the plates.

Figure 4 (b) shows the calculation results of the initial water level distribution of the tsunami using the Okada (1985) equation. The initial water level of up to approximately + 3.5 m is distributed off to Sagami Bay and Tokyo Bay. Using the initial water level as an input value, we performed a tsunami numerical simulation via a nonlinear long wave equation. We use the following continuity equation (Eq. (13)) and nonlinear shallow water equations (Eqs. (14) and (15)) as follows:

$$\frac{\partial \eta}{\partial t} + \frac{\partial M}{\partial x} + \frac{\partial N}{\partial y} = 0 \tag{13}$$

$$\frac{\partial M}{\partial t} + \frac{\partial}{\partial x}\left[\frac{M^2}{D}\right] + \frac{\partial}{\partial y}\left[\frac{MN}{D}\right] + gD\frac{\partial \eta}{\partial x} + \frac{gn^2}{D^{7/3}}M\sqrt{M^2 + N^2} = 0 \tag{14}$$

$$\frac{\partial N}{\partial t} + \frac{\partial}{\partial x}\left[\frac{MN}{D}\right] + \frac{\partial}{\partial y}\left[\frac{N^2}{D}\right] + gD\frac{\partial \eta}{\partial y} + \frac{gn^2}{D^{7/3}}N\sqrt{M^2 + N^2} = 0 \tag{15}$$

where $\eta$ denotes the water level, $D$ denotes the total water level, $g$ denotes the acceleration due to gravity, $n$ denotes the Manning coefficient, and $M$ and $N$ denote the fluxes in the $x$ and $y$ directions, respectively. The governing equations were discretized via the staggered leapfrog scheme (Goto and Ogawa, 1982; UNESCO, 1997). To consider wave height uncertainty, we implemented 25 cases of tsunami numerical simulation for each earthquake source. As detailed in the second chapter, this study focused on the slip amount and the fault depth among many uncertain factors. In each source, the slip amount was varied by ± 0.1 times, ± 0.05 times with respect to the reference case (5 cases) in terms of Mw conversion based on the scaling law, and the fault depth was changed by + 2.0 km, +1.0 km, - 0.5 km, and - 1.0 km with respect to the reference case (5 cases) to consider the changes of the slip and the fault depth as uncertainty.

There are a total of 10 earthquake sources thus, we implemented a total of 250 cases of tsunami numerical simulation nested in four stages of 270 m, 90 m, 30 m, and 10 m in the Japanese plane rectangular coordinate system IX for each simulation and executed the simulation for 3 hours from the earthquake occurrence. As an example, Fig. 5 shows the numerical simulation results of 9 cases around Oiso and Miura in which the Mw of the source 8 is changed to ± 0.1, the fault depth is changed to + 2.0 km, and - 1.0 km. As shown in the figure, the distributions of the maximum tsunami wave height vary locally by changing the slip amount and the fault depth, and the effect of the slip amount on the maximum tsunami wave height is more dominant than the fault depth. In addition, while there is a clear positive correlation between the maximum tsunami wave height and slip amount of the earthquake, there is no clear correlation between the maximum tsunami wave height and the fault depth. Figure 6 shows the maximum tsunami wave heights of Miura and Oiso and Pearson's correlation coefficient relative to the tsunami numerical simulation results of each earthquake source. We confirmed that the correlation coefficient corresponded to at least 0.8 in any source, thus the correlation between tsunami wave height of Miura and Oiso was relatively high. The results suggest that we should assess tsunami risk considering the spatial correlation of tsunami wave height between the target points.

### 3.3 Construction of response surface

In this section, we construct response surfaces, which indicate maximum wave height at target sites. With respect to the results of the maximum wave height of the tsunami numerical simulation, we regressed the response surface (Eq. (2)) using the least squares method. The explanatory variables correspond to the fault slip and the fault depth, and the

objective variable denotes the maximum wave height at the target sites. We performed the regression analysis based on all combinations of four explanatory variables ($2^4 - 1 = 15$ cases) and adopted a response surface with a high coefficient of determination and the minimum Akaike Information Criterion (AIC) (Akaike, 1974). AIC can compare the quality of a set of statistical models to each other. The best model is the one that has the minimum AIC among all the other models. Table 3 shows the AIC values of 15 case regression analyses for Miura and Oiso, and Table 4 shows the regression coefficients of the response surface where AIC corresponds to the minimum in each earthquake source. For example, Figs. 7 (a) (b) shows the response surface for the earthquake source 8 (Mw: 8.6) with the highest Mw in the Sagami Trough earthquake. The blue circle denotes the maximum wave height obtained from the tsunami numerical simulations, and the red curved surface denotes the response surface. The response surfaces accurately represented the results of the tsunami numerical simulation. The response surfaces are in accordance with Eq. (16) for Oiso and Eq. (17) for Miura as follows:

$$h(S, D) = 0.6567S + 0.0459D - 0.5189S^2 + 0.5147 \tag{16}$$

$$h(S, D) = 11.1136S - 4.0165S^2 - 3.1327 \tag{17}$$

We can obtain the frequency distribution of the tsunami wave height by giving a probability distribution function that expresses the uncertainty to the explanatory variable (slip ratio $S$ and fault depth $D$) of the evaluated response surface and performing a Monte-Carlo simulation.

As reported by JSCE (2002), the estimated variation of Mw of an earthquake of the same magnitude is approximately 0.1. Based on the aforementioned value, we set a normal distribution with an average value of 1.0 and a standard deviation of 0.1 for the slip rate by using the scaling law. With respect to the uncertainty of the fault depth, we also set a normal distribution. The average value was set to 0.0 m, and the standard deviation was set to a random number generated from a log normal distribution that was obtained from the seismic observation error data from October 2016 to September 2017 (N = 305,030) as published by the Japan Meteorological Agency (2017). We used the lognormal distribution with an average of 0.12 km and a standard deviation of 0.65 km. We would like to note that it is necessary essentially to apply a probability distribution that appropriately expresses all possible uncertainties to the explanatory variables of the response surface, but in this study we applied a relatively limited probability distribution as uncertainty since we did not focus on discussing the details of the tsunami wave uncertainty, but on proposed tsunami hazard and risk assessment method using response surface and copulas. Figures 8 (a) (b) shows the frequency distribution of the tsunami wave height obtained by the aforementioned procedure. By using the response surface method, we can significantly reduce the simulation costs for probabilistic tsunami hazard assessment considering uncertainty.

To ascertain the normality of the frequency distributions, we performed the Kolmogorov–Smirnov test. Table 5 shows the results of p-values for each source. In several cases the p-values were less than 0.05, thereby indicating that the distribution of the tsunami heights does not necessarily follow a normal distribution.

### 3.4 Dependence modelling using copulas

In this section, we estimate appropriate copulas from the results of tsunami numerical simulation considering uncertainties and evaluate spatial correlation structure of tsunami wave height between two sites.

As confirmed in the previous section, despite the high linear correlation of the frequency distribution of the tsunami wave height in Miura and Oiso, it is observed that the normality of tsunami wave height for several sources was not secured by the normality test. The Pearson's correlation coefficient did not accurately grasp the spatial correlation structure of tsunami wave height, and thus we attempt modeling using copula. Hereafter, we only illustrate the analysis results of the earthquake source 8 (Mw: 8.6) with the largest Mw as an example.

Table 6 shows the results of estimating copulas by maximum likelihood estimation for the distribution obtained via converting the numerical simulation results over [0, 1]. We considered that a copula associated with the minimum AIC and Bayesian Information Criterion (BIC) (Schwarz, 1978) as the best-fit copula. The BIC is more useful in selecting a correct model while the AIC is more appropriate in finding the best model for predicting future observations. In source 8, the copula with the minimum AIC and BIC corresponded to the Frank copula. We derived the joint distribution of the tsunami wave heights considering the wave height correlation using the Frank copula and the empirical cumulative distributions obtained from the histogram of the tsunami wave height evaluated in the previous section. Figure 9 shows the Frank copula over [0, 1] with 10,000 trials, Figs. 10 (a) and (b) show the empirical cumulative distributions of tsunami wave height for Oiso and Miura, Fig. 11 (a) shows the results considering the wave height correlation. The black points denote the results of Monte-Carlo simulation. The number of simulations is 10,000. The red points denote the results of tsunami numerical simulation by nonlinear long wave equation. To compare with this result, Fig. 11 (b) shows the results without considering the wave height correlation. We independently generated the tsunami wave height by using a uniform random number and the cumulative frequency distribution of the tsunami wave height at each site without using a copula. By considering the spatial correlation of the tsunami wave heights using copula, we performed a Monte-Carlo simulation that appropriately captures the nonlinear spatial correlation of the tsunami wave height. We clearly showed the usefulness of copula modeling considering the wave height correlation.

Table 7 shows the result of estimating copulas under the same procedure for other earthquake sources. In the earthquake sources targeted in this study, four types of copula were estimated, namely rotated Gumbel copula, asymmetric Gumbel copula, Frank copula, and Gumbel copula. Rotated Gumbel copula corresponds to a copula that rotates the ordinary Gumbel copula by 180 degrees. For reference purposes, the copulas for all earthquake sources are illustrated in Fig. 12. From the characteristics of the copula mentioned before, there is a tail dependency in the wave heights due to the source 1, 2, 3, 5, 7 and 9, but there is no tail dependency in the wave heights due to the source 4, 6, 8 and 10. The tail dependency of the wave height could change in various ways under the effects from the relative position of the earthquake sources and the target points, the bottom and land topography.

### 3.5 Risk assessment results and discussion

In this section, we evaluate the joint distribution of tsunami wave heights and damage probability of target buildings for the entire area of the Sagami Trough earthquake using the occurrence probability weights of each earthquake source.

Table 8 shows the occurrence probability weights of each source of the Sagami Trough earthquake published by NIED (2017). We first determine the earthquake occurrence source via uniform random numbers using the weights and then evaluate the joint distribution of the tsunami wave heights due to the determined earthquake using the estimated copula. Figure 13 shows the results of evaluation by Monte-Carlo simulation with 10,000 trials. Figure 13 (a) shows the joint distribution of the tsunami wave heights considering the spatial correlation of the wave height, and Fig. 13 (b) shows the results without considering spatial correlation of the tsunami wave height. Furthermore, Fig. 13 (c) shows the joint damage probability of two buildings that transform both axes of tsunami wave heights in Fig. 13 (b) into the damage probability by using the damage function of the steel frame (Suppasri et al., 2013) based on the assumption that a steel building exists at the evaluation target point. Table 9 shows the average value of aggregate damage probability of two buildings, 95 percentile value, 99 percentile value, and maximum value assuming that the two buildings exhibit the same asset value. Although the expected value of the aggregate damage probability barely changed when compared with that of no correlation case, the aggregate damage probability of the ninety-ninth percentile value was approximately 1.0 % higher and the maximum value was approximately 3.0 % higher when considering the hazard correlation utilizing the copulas. We clearly showed the significance of considering spatial correlation structure of tsunami wave height in evaluating tsunami risks for a building portfolio. In this study we only demonstrated the evaluation method for two points, but the effect of the wave height correlation on the evaluation result is expected to increase if more points are targeted.

### 4 Conclusion

In this study, we evaluated the aggregate tsunami damage probability of two buildings located at two relatively remote locations based on the frequency distribution of the tsunami height via the response surface method and the spatial correlation of the tsunami height by using various copulas, assuming the occurrence of the Sagami Trough earthquake that significantly affects the metropolitan area in Japan. The ninety-ninth percentile value of the aggregate damage probability was approximately 1.0 % higher, and the maximum value was approximately 3.0 % higher in the evaluation considering the spatial correlation of the tsunami wave height when compared with the evaluation without considering the spatial correlation. The results clearly show the significance of considering the spatial correlation of the tsunami hazard in evaluating tsunami risks for a building portfolio and suggest that spatial correlation modeling by copulas is effective in the case wherein nonlinear correlation of the tsunami hazard exists. In addition, the response surface method used in this study significantly reduces the numerical simulation costs for probabilistic tsunami hazard assessment considering uncertainty. In this study, we only focused on the slip amount and fault depth among many tsunami hazard uncertainties, and evaluated them using the response surface method. It has been

reported that the heterogeneity of the slip distribution of the fault has a great influence on tsunami intensity. It is a future issue to evaluate these effects with a response surface method.

The evaluation result was shown for only two buildings, but when an entity evaluates the risk of assets it owns it is assumed that there will be more target sites. It is clear that as the number of target assets increases, the percentile value and maximum value of aggregate damage of assets becomes more prominent. Risk assessment that does not consider the spatial correlation of wave heights will lead to underestimation of the risks held. The basic method shown in this study can be applied even when the number of target assets increases. It is also important to avoid underestimating the assessed risk by considering the wave height correlation using a copula. It is expected that the tsunami risk assessment method for a building portfolio over a wide area as proposed in this study can be used for probabilistic tsunami risk assessment of real estate portfolios or business continuity plans by parties such as large companies, insurance companies and real-estate agencies.

## Author contributions

YF conceived and designed the experiments, analyzed the data and wrote the paper with assistance and input from SM, KT, TK, YO and TK.

## Competing interests

The authors declare that they have no conflict of interest.

## Acknowledgement

We thank two reviewers who provided us valuable comments and helped improve the manuscript. This research was partially supported by funding from the International Research Institute of Disaster Science (IRIDeS) at Tohoku University.

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

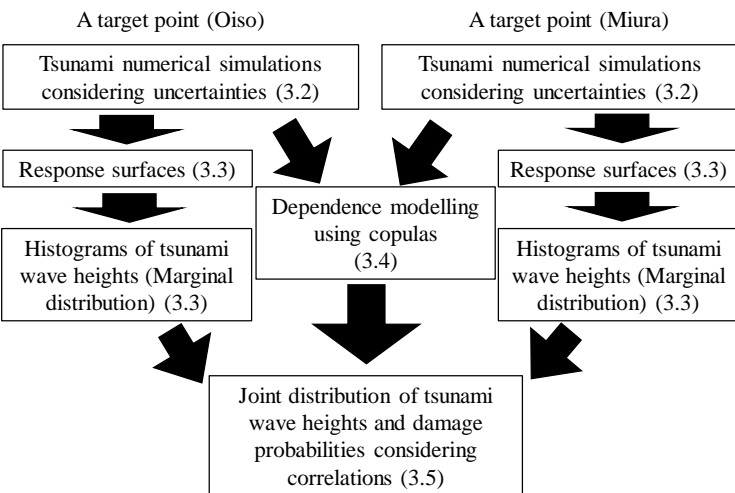

**Figure 1: Flowchart of probabilistic tsunami hazard and risk assessment considering spatial correlation of tsunami wave height. Numbers in the parentheses indicate the section numbers escribed.**

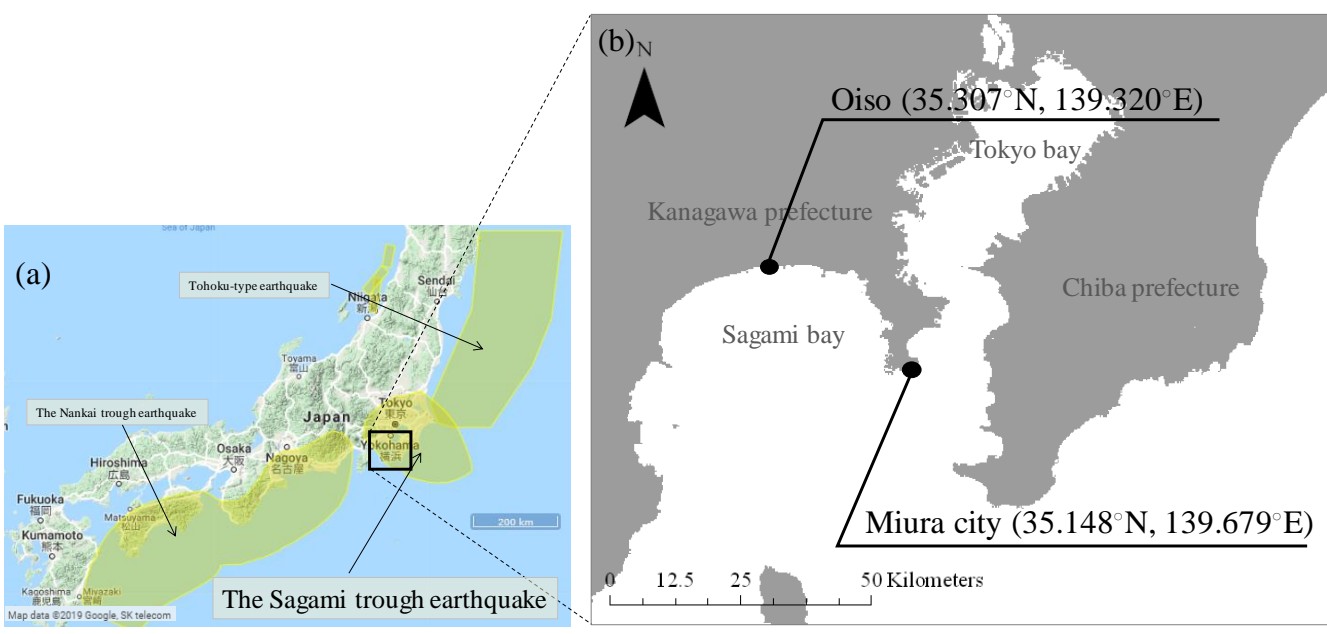

**Figure 2: (a) Major subduction-zone earthquakes around Japan islands including the Sagami Trough earthquake, the Nankai Trough earthquake and Tohoku-type earthquake (Yellow area), (b) Two targets points, Oiso and Miura, Kanagawa prefecture for tsunami hazard and risk assessment**

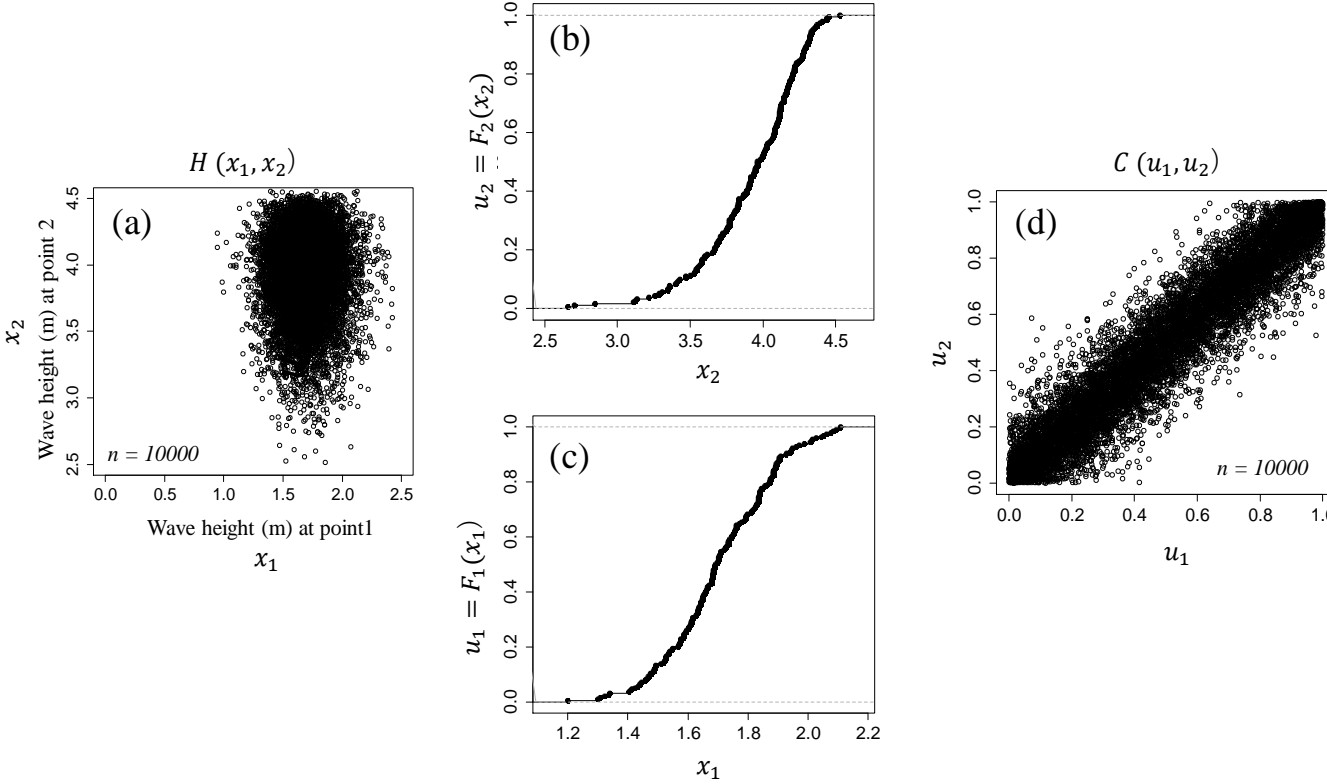

Figure 3: A simple synthetic example of a copula in bivariate case. (a) Joint distribution, (b) and (c) are distribution functions of each variable (marginal distribution) and (c) is a copula distributed over [0, 1].

Table 1: Bivariate copula, parameter vectors, Kendall's tau

| Copula | Parameter vectors | Kendall's tau |
|---|---|---|
| Gaussian copula | $\rho$ | $(2/\pi)\arcsin\rho$ |
| t copula | $\rho, v$ | $(2/\pi)\arcsin\rho$ |
| Clayton copula | $\theta$ | $\theta/(\theta+2)$ |
| Frank copula | $\theta$ | $1-4/\theta+4D_1(\theta)/\theta$ |
| Gumbel copula | $\theta$ | $1-1/\theta$ |
| Asymmetric Gumbel copula | $r, \theta, \varphi$ | $\int_0^1 \frac{t(1-t)A''(t)}{A(t)} dt$ |

$\rho$: Pearson's correlation coefficient, $D_1(\theta) = \int_0^\theta \frac{\frac{x}{\theta}}{(e^x-1)} dx$ : the first Debye function

**Table 2: Moment magnitude, Average slip, Number of faults and Area in each earthquake source of the Sagami Trough earthquake**

| Source number | Moment magnitude (Mw) | Average slip (m) | Number of faults | Area (km$^2$) |
|---|---|---|---|---|
| 1 | 7.9 | 2.5 | 1207 | 7544 |
| 2 | 8.2 | 4.0 | 2392 | 14950 |
| 3 | 8.0 | 2.7 | 1533 | 9581 |
| 4 | 8.3 | 4.6 | 3393 | 21206 |
| 5 | 8.4 | 5.0 | 3599 | 22494 |
| 6 | 8.5 | 5.8 | 4926 | 30788 |
| 7 | 8.5 | 5.2 | 4822 | 30138 |
| 8 | 8.6 | 6.3 | 6149 | 38431 |
| 9 | 7.9 | 2.5 | 1234 | 7713 |
| 10 | 8.2 | 3.0 | 2825 | 17656 |

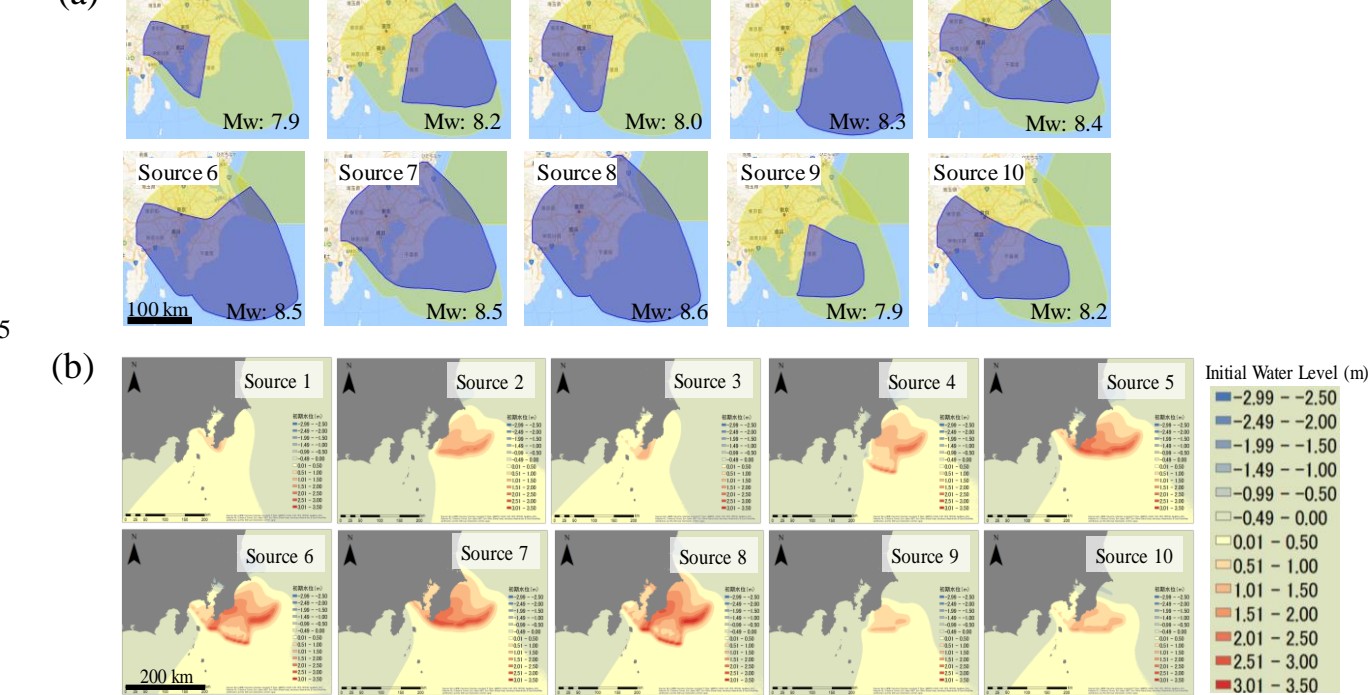

**Figure 4: (a) The ten sources of the Sagami Trough earthquakes (NIED, 2017) and (b) Initial water levels of tsunami calculated from the fault parameters using Okada equation (Okada, 1985).**

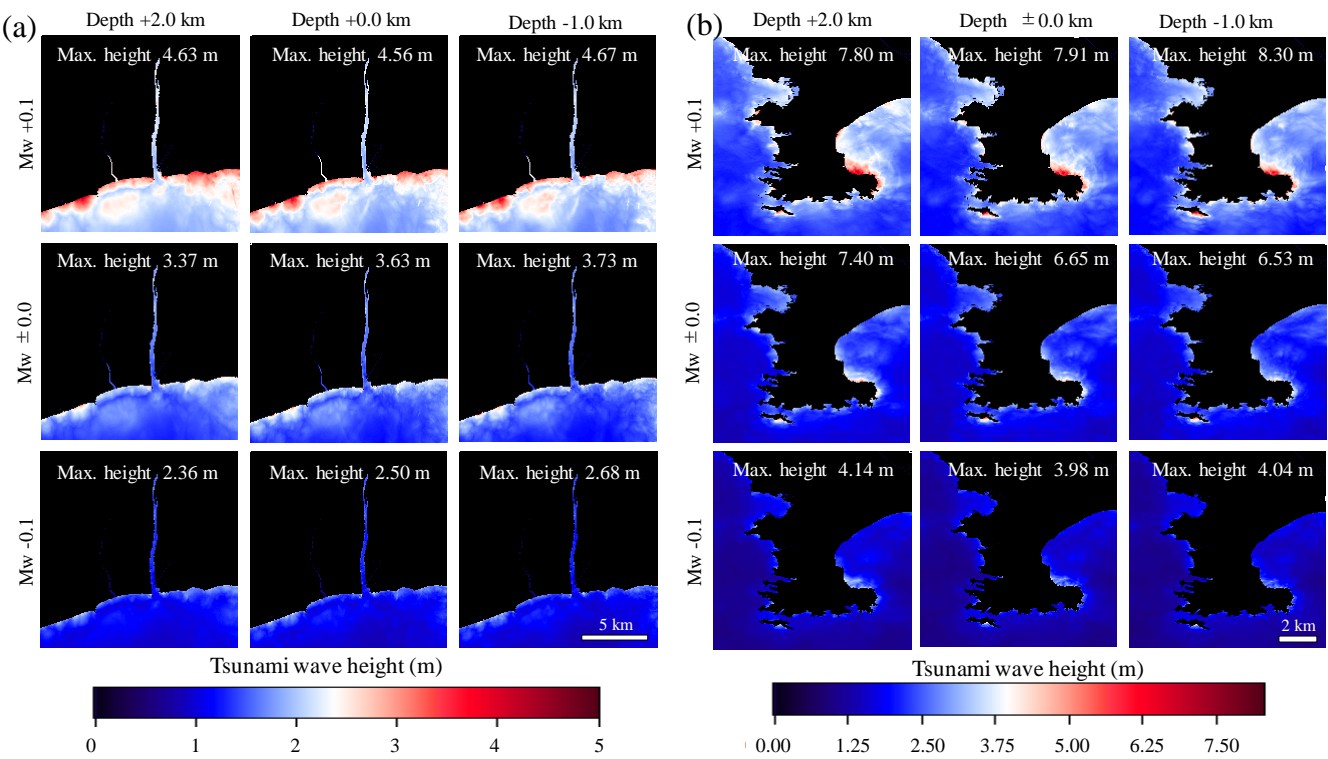

**Figure 5: Tsunami numerical simulation results ((a) Oiso and (b) Miura) in case of changing the Mw (Moment Magnitude) and the fault depth of the source 8**

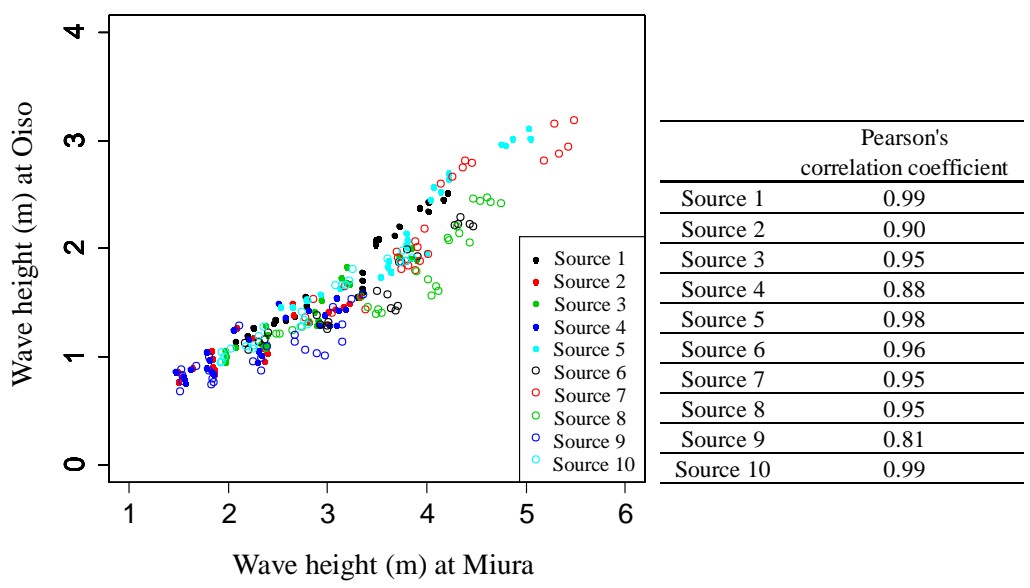

| | Pearson's correlation coefficient |
|---|---|
| Source 1 | 0.99 |
| Source 2 | 0.90 |
| Source 3 | 0.95 |
| Source 4 | 0.88 |
| Source 5 | 0.98 |
| Source 6 | 0.96 |
| Source 7 | 0.95 |
| Source 8 | 0.95 |
| Source 9 | 0.81 |
| Source 10 | 0.99 |

**Figure 6: Maximum tsunami wave heights simulated from tsunami numerical simulation at Miura and Oiso and Pearson's correlation coefficients in each earthquake source.**

**Table 3: Akaike Information Criterion (AIC) results of the regression analyses.**

**The regression analysis were performed based on all combinations of four explanatory variables.**

| Oiso | Cases of the response surface | | | | | | | | | | | | | | | |
|---|---|---|---|---|---|---|---|---|---|---|---|---|---|---|---|---|
| | 1 | 2 | 3 | 4 | 5 | 6 | 7 | 8 | 9 | 10 | 11 | 12 | 13 | 14 | 15 | Minimum |
| Source 1 | -86.1073 | -75.0959 | -76.4418 | -86.0675 | -83.3907 | -69.8912 | -47.8628 | -83.6900 | 37.3278 | -76.0403 | -75.3817 | -46.9466 | -48.9226 | 36.8890 | 36.3744 | -86.1073 |
| Source 2 | -70.4205 | -62.1961 | -66.0866 | -71.2517 | -71.4697 | -60.0875 | -47.2552 | -72.3436 | 8.2820 | -63.5425 | -67.3455 | -45.3779 | -48.9304 | 7.5681 | 6.9034 | -72.3436 |
| Source 3 | -78.1960 | -69.4144 | -73.0031 | -78.9561 | -79.9437 | -66.8035 | -51.6857 | -80.7160 | 24.0401 | -70.7359 | -74.8137 | -50.0483 | -53.6110 | 23.5389 | 22.9902 | -80.7160 |
| Source 4 | -70.3635 | -62.7432 | -67.6975 | -72.0181 | -71.4781 | -61.7123 | -46.0815 | -73.1446 | 6.5531 | -64.5753 | -68.9606 | -44.5848 | -47.7924 | 6.0516 | 5.2790 | -73.1446 |
| Source 5 | -84.3052 | -79.8555 | -83.1754 | -86.2150 | -70.7497 | -79.5811 | -51.9906 | -72.7012 | 45.9869 | -81.7477 | -71.0212 | -52.3149 | -49.3924 | 45.8946 | 45.4242 | -86.2150 |
| Source 6 | -84.3409 | -85.3398 | -81.8169 | -85.7067 | -71.5073 | -83.0709 | -64.9049 | -73.1549 | 31.6107 | -86.7700 | -70.9075 | -66.5245 | -60.0166 | 31.2625 | 30.8593 | -86.7700 |
| Source 7 | -21.7317 | -18.3202 | -23.5936 | -23.7244 | -23.6513 | -20.2440 | -22.2107 | -25.6440 | 48.8843 | -20.3017 | -25.5136 | -19.4355 | -24.1409 | 48.7116 | 48.3525 | -25.6440 |
| Source 8 | -81.1962 | -79.0259 | -77.2264 | -82.0455 | -73.1696 | -76.0809 | -59.6378 | -74.3933 | 35.9058 | -80.1470 | -71.0190 | -60.0970 | -57.5528 | 35.5037 | 35.0967 | -82.0455 |
| Source 9 | -31.0739 | -32.3196 | -31.8511 | -32.9766 | -29.8352 | -33.0836 | -25.1204 | -31.7497 | 4.7047 | -34.2103 | -30.7579 | -26.6012 | -24.8998 | 3.8330 | 3.1376 | -34.2103 |
| Source 10 | -80.2635 | -69.9115 | -73.6468 | -80.1713 | -82.1713 | -66.6323 | -53.8587 | -82.0864 | 23.4459 | -70.7917 | -75.5814 | -51.5993 | -55.8313 | 22.8583 | 22.3445 | -82.1713 |

| Miura | Cases of the response surface | | | | | | | | | | | | | | | |
|---|---|---|---|---|---|---|---|---|---|---|---|---|---|---|---|---|
| | 1 | 2 | 3 | 4 | 5 | 6 | 7 | 8 | 9 | 10 | 11 | 12 | 13 | 14 | 15 | Minimum |
| Source 1 | -29.9142 | -3.7669 | -29.5662 | -31.5773 | -18.4609 | -5.1563 | -19.9724 | -20.2637 | 56.8927 | -5.7181 | -19.0634 | -2.3947 | -13.4162 | 56.4377 | 55.9832 | -31.5773 |
| Source 2 | -32.4696 | -21.1006 | -34.4497 | -34.1257 | -32.7950 | -23.1000 | -32.9901 | -34.4733 | 51.9175 | -22.9686 | -34.7765 | -23.0689 | -33.5269 | 51.4723 | 51.5649 | -34.7765 |
| Source 3 | -43.3459 | -30.2321 | -43.9870 | -43.9721 | -44.8801 | -31.6225 | -45.9601 | -45.5310 | 55.2425 | -31.6189 | -45.5455 | -33.6121 | -47.5191 | 54.6938 | 54.6028 | -47.5191 |
| Source 4 | -22.4764 | -12.7328 | -23.4804 | -21.3515 | -22.0638 | -14.2238 | -15.3526 | -21.2106 | 50.6507 | -12.9471 | -23.1577 | -9.3945 | -15.7799 | 49.5413 | 49.8184 | -23.4804 |
| Source 5 | -3.5315 | 1.5932 | -4.8179 | -5.3684 | -4.4497 | 0.0418 | -3.8392 | -6.2935 | 58.8979 | -0.3264 | -5.7659 | 0.3814 | -4.9032 | 58.3657 | 58.0118 | -6.2935 |
| Source 6 | -16.9265 | 8.9520 | -18.6108 | -18.1964 | 7.9340 | 7.0088 | -20.5546 | -5.0276 | 61.2059 | 6.9971 | -5.0028 | 5.0157 | -6.9975 | 60.6742 | 60.5649 | -20.5546 |
| Source 7 | 3.3372 | 1.3587 | 2.1765 | 1.5142 | 3.7790 | 0.1910 | 3.8906 | 1.9395 | 63.6719 | -0.4676 | 2.5413 | 1.9058 | 3.9420 | 63.0994 | 62.7084 | -0.4676 |
| Source 8 | -27.0282 | 19.0202 | -27.1925 | -26.7027 | 7.9340 | 17.1854 | -28.3906 | 6.4835 | 60.0863 | 17.2428 | 6.3645 | 15.2927 | 4.5602 | 59.1621 | 59.1455 | -28.3906 |
| Source 9 | -34.2223 | -26.1205 | -36.1871 | -36.0073 | -35.9841 | -28.1145 | -36.7777 | -37.7711 | 52.5198 | -28.0294 | -37.9493 | -29.1978 | -38.5528 | 52.1439 | 52.1581 | -38.5528 |
| Source 10 | -55.5283 | -42.4949 | -53.1771 | -54.3099 | -57.3486 | -42.2671 | -54.1155 | -56.1518 | 55.2739 | -42.9033 | -55.0260 | -43.5964 | -55.9706 | 54.6558 | 54.4912 | -57.3486 |

**Table 4: Regression coefficients of selected each response surface for each earthquake source**

| Oiso | Regression coefficients | | | | |
|---|---|---|---|---|---|
| | a | b | c | d | e |
| Source 1 | 1.1705 | 0.1039 | -0.0371 | 0.3051 | 0.1927 |
| Source 2 | 0.9868 | 0.0598 | 0.0000 | 0.0000 | 0.1037 |
| Source 3 | 1.3747 | 0.0566 | 0.0000 | 0.0000 | 0.0040 |
| Source 4 | 0.9568 | 0.0625 | 0.0000 | 0.0000 | 0.1184 |
| Source 5 | 0.7991 | 0.0592 | 0.0000 | 0.6449 | 0.6303 |
| Source 6 | 0.0000 | 0.0404 | 0.0000 | 0.7610 | 0.7538 |
| Source 7 | 2.2360 | 0.0445 | 0.0000 | 0.0000 | -0.0971 |
| Source 8 | 0.6567 | 0.0459 | 0.0000 | 0.5189 | 0.5147 |
| Source 9 | 0.0000 | 0.0661 | 0.0000 | 0.3945 | 0.5739 |
| Source 10 | -1.3690 | -0.0972 | 0.0423 | 0.0000 | -0.0029 |

| Miura | Regression coefficients | | | | |
|---|---|---|---|---|---|
| | a | b | c | d | e |
| Source 1 | 6.2764 | 0.0832 | 0.0000 | -1.7394 | -1.3700 |
| Source 2 | 2.3946 | 0.0000 | -0.0336 | 0.0000 | -0.1281 |
| Source 3 | 2.5601 | 0.0000 | 0.0000 | 0.0000 | 0.2187 |
| Source 4 | 3.8893 | 0.0000 | -0.0767 | -0.7610 | -0.8384 |
| Source 5 | 2.6802 | 0.0643 | 0.0000 | 0.0000 | 1.0744 |
| Source 6 | 8.0738 | 0.0000 | 0.0000 | -2.5004 | -2.1023 |
| Source 7 | 0.0000 | 0.0829 | 0.0000 | 1.3910 | 2.4982 |
| Source 8 | 11.1136 | 0.0000 | 0.0000 | -4.0165 | -3.1327 |
| Source 9 | 2.4222 | 0.0000 | 0.0000 | 0.0000 | -0.1673 |
| Source 10 | -2.5917 | -0.1083 | 0.0869 | 0.0000 | -0.1061 |

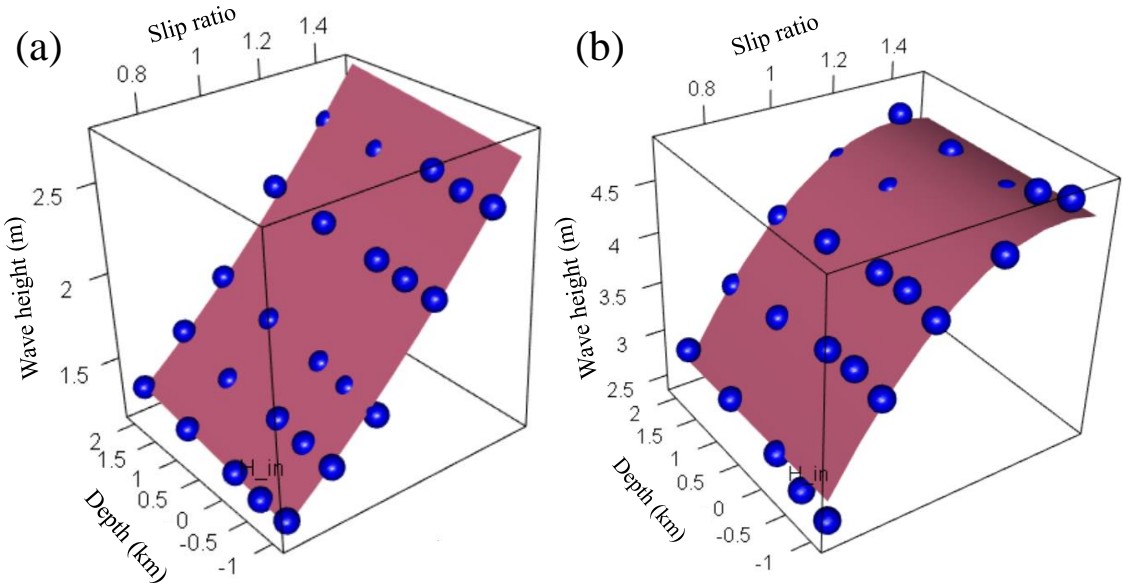

**Figure 7: Response surfaces at (a) Oiso and (b) Miura for the source 8 of the Sagami Trough earthquake. The blue circle denotes the maximum wave height obtained from the tsunami numerical simulations and the red curved surface denotes the response surface.**

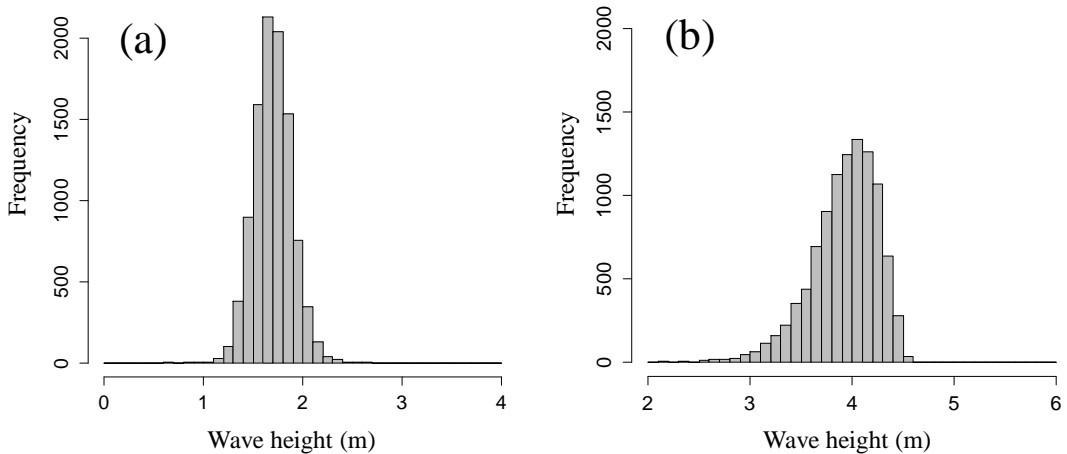

**Figure 8: Histograms of tsunami wave height simulated from the response surface at (a) Oiso and (b) Miura for the source 8 of the Sagami Trough earthquake**

**Table 5: Kolmogorov–Smirnov test results**

|  | p-value | |
| --- | --- | --- |
|  | Oiso | Miura |
| Source 1 | 0.00 | 0.00 |
| Source 2 | 0.00 | 0.89 |
| Source 3 | 0.00 | 0.61 |
| Source 4 | 0.00 | 0.15 |
| Source 5 | 0.07 | 0.95 |
| Source 6 | 0.72 | 0.02 |
| Source 7 | 0.79 | 0.50 |
| Source 8 | 0.26 | 0.00 |
| Source 9 | 0.00 | 0.93 |
| Source 10 | 0.03 | 0.97 |

**Table 6: Maximum likelihood estimation results of each copulas for the source 8**

| Name of copulas | Log-likelihood | AIC | BIC |
| --- | --- | --- | --- |
| Gaussian copula | 24.72 | -47.43 | -46.21 |
| t-copula | 24.62 | -45.23 | -42.79 |
| Clayton copula | 24.46 | -46.93 | -45.71 |
| Gumbel copula | 20.03 | -38.06 | -36.84 |
| Frank copula | 26.16 | -50.33 | -49.11 |
| rotated Clayton copula | 14.53 | -27.06 | -25.84 |
| rotated Gumbel copula | 25.77 | -49.54 | -48.32 |
| asymmetric Gumbel copula | 19.90 | -35.80 | -33.36 |
| rotated Asymmetric Gumbel copula | 25.69 | -47.38 | -44.94 |

Figure 9: Selected Frank copula for the source 8

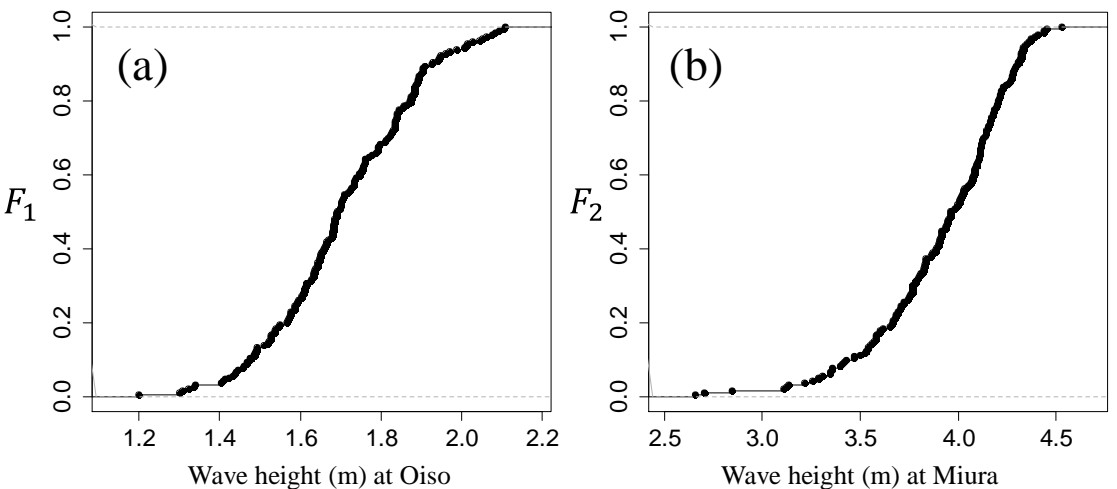

**Figure 10: Empirical cumulative distributions of tsunami wave height ((a) Oiso and (b) Miura) for the source 8**

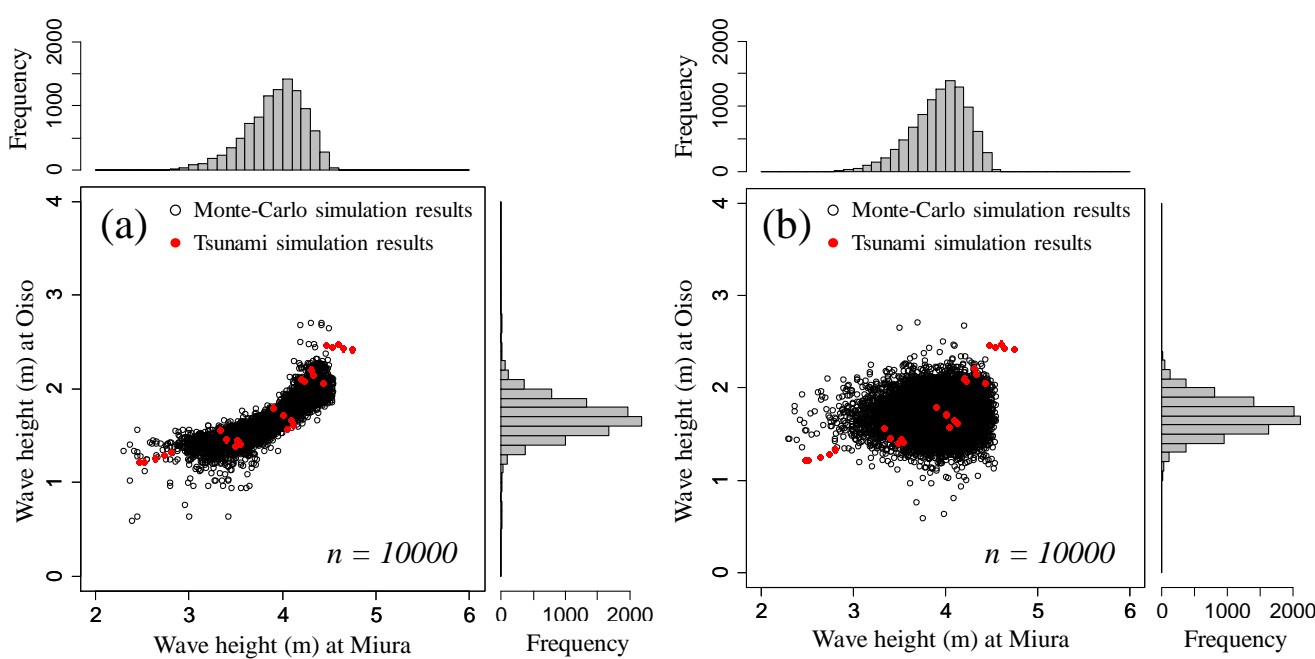

**Figure 11: Monte-Carlo simulation results for the source 8. The black points denote the results with 10,000 trials (a) considering and (b) not considering the spatial correlation of tsunami wave heights using the Frank copulas. The red points denote the results calculated from 25 cases of tsunami numerical simulation.**

**Table 7: Estimated optimal copulas, copula parameters, and Kendall's tau for each source of the Sagami Trough earthquake**

|  | Estimated copulas | Parameters | Kendall's $\tau$ |
|---|---|---|---|
| Source 1 | rotated Gumbel copula | 20.42 | 0.95 |
| Source 2 | asymmetric Gumbel copula | 1.00, 5.08, 0.85 | 0.70 |
| Source 3 | rotated Gumbel copula | 4.62 | 0.78 |
| Source 4 | Frank copula | 10.54 | 0.68 |
| Source 5 | rotated Gumbel copula | 9.24 | 0.89 |
| Source 6 | Frank copula | 22.11 | 0.83 |
| Source 7 | Gumbel copula | 5.68 | 0.82 |
| Source 8 | Frank copula | 17.77 | 0.80 |
| Source 9 | Gumbel copula | 2.87 | 0.65 |
| Source 10 | Frank copula | 35.76 | 0.89 |

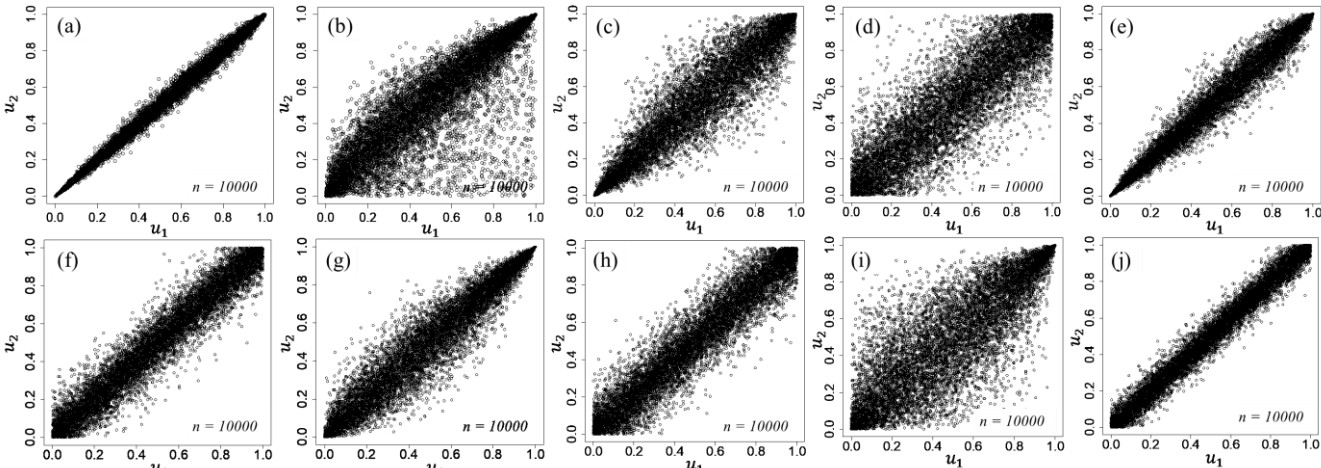

**Figure 12: Estimated optimal copulas distributed on $[0, 1]^2$ with 10,000 trials. (a) rotated Gumbel copula for the source 1, (b) asymmetric Gumbel copula for the source 2, (c) rotated Gumbel copula for the source 3, (d) Frank copula for the source 4, (e) rotated Gumbel copula for the source 5, (f) Frank copula for the source 6, (g) Gumbel copula for the source 7, (h) Frank copula for the source 8, (i) Gumbel copula for the source 9, (j) Frank copula for the source 10**

**Table 8: Occurrence probability weights of each source of the Sagami Trough earthquake (NIED, 2017)**

|  | Occurrence probability weights |
|---|---|
| Source 1 | 0.37 |
| Source 2 | 0.06 |
| Source 3 | 0.30 |
| Source 4 | 0.05 |
| Source 5 | 0.03 |
| Source 6 | 0.01 |
| Source 7 | 0.01 |
| Source 8 | 0.02 |
| Source 9 | 0.11 |
| Source 10 | 0.04 |
| Summation | 1.00 |

**Table 9: Tsunami risk assessment results**

|  | Aggregate damage probability of two buildings | | |
|---|---|---|---|
|  | No correlation (A) | Correlation (B) | Difference (B-A) |
| Average | 58.8% | 58.8% | 0.0% |
| 95 percentile | 66.2% | 67.0% | 0.9% |
| 99 percentile | 68.9% | 69.7% | 0.8% |
| Maximum | 73.5% | 76.7% | 3.1% |

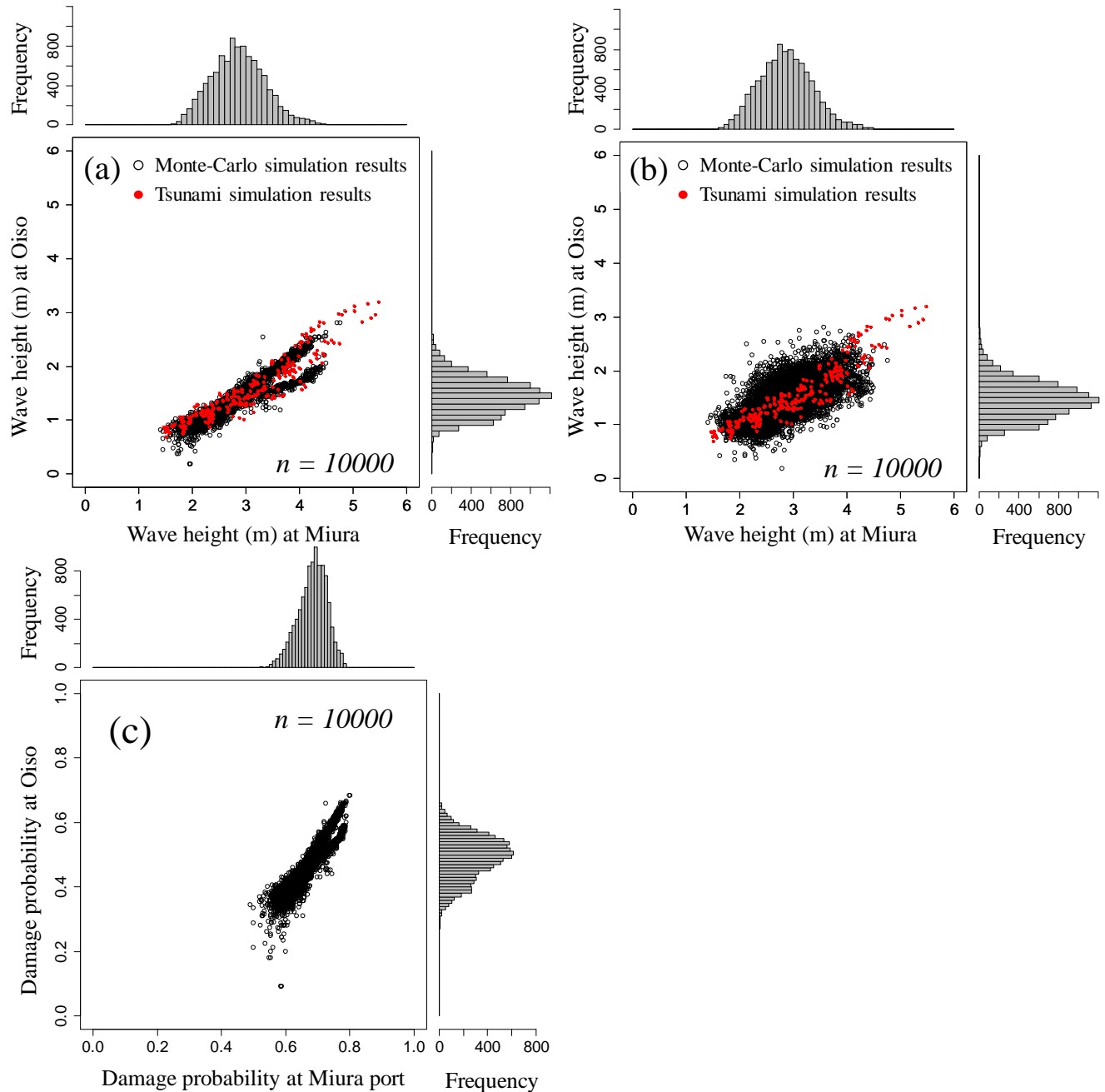

**Figure 13:** **(a) Joint distribution of tsunami wave height considering wave height correlation (b) not considering wave height correlation and (c) Joint damage probability for the all sources of the Sagami Trough earthquake. The black points denote the Monte-Carlo simulation results with 10,000 trials and the red points denote the results simulated via tsunami numerical simulations.**

