# Peer review of "Tsunami hazard and risk assessment for multiple buildings by considering spatial correlation of wave height using copulas"

_Natural Hazards and Earth System Sciences, 2019_

## Referee Comment (RC1) · Anonymous Referee #1 · 26 Jun 2019

This paper presents a new statistical method for relating the hazard and risk at different locations due to the same scenario. From my reading of the paper, it seems like the main emphasis is to use spatial correlation methods in order to reduce the computational burden in tsunami hazard and risk studies, in particular with respect to computing local hazard maps. The method is unconventional, and there few or no similar studies in the tsunami literature of this kind, which makes the study a nice addition to the literature. However, the explanation of the methodology and results in order for other practitioner to utilize the findings in other situations is not fully clear. In several places in the paper, explanations are too brief, and sometimes key concepts are not explained. Yet, the paper might be considered appropriate for publication given that the following

items are improved and clarified:

- The description of copula methodology should be improved and elaborated. Essential characteristics of the methods needs to be spelled out in more details, especially in order to make the methodology transparent for NHESS readers that are not experts in probability theory and copulas. For instance, key concepts such as the transformed variable u is not even defined. Source mechanisms must be described better and in the full detail necessary.

- Limitations related with the methodology and results (i.e. the cases investigated) should be better clarified, both with a discussion, but also with some further quantification of uncertainties that are suppressed in the present version of the paper.

- The application of the method presented in the final part of the paper is key to understand the impact of the method, as it demonstrates that the coupled probability is necessary for understanding interspatial correlation. However, the description of the Monte Carlo type realisation is too brief. The sampling methods should be explained in more detail. I take it that the non-correlated results are simply sampled randomly from both marginal distributions, but this is not explained in sufficient detail anywhere.

-It is not described anywhere how the method can be used in probabilistic risk or hazard assessments, despite the fact that this is stressed in the motivation for the paper. It would be interesting if the authors could provide more details on how the copula methodology could be exploited using probabilistic methods.

More details are given in the line-by-line comments.

Line by line comments

Page 1-line 24: It would preferable if key concepts of probabilistic tsunami hazard assessment (PTHA) and probabilistic tsunami risk assessment are introduced here. The authors are simply referring to them without explaining what they are. Some more details on PTHA and PTRA would be preferable.

Page 1 – line 25: This study is very much on hazard as well as the risk. I suggest including hazard also in the title. Replace "the probabilistic risk..." with "a variety of probabilistic hazard and risk..."

Page 1 – line 28: Relevant overview that preferably should be added to these references would be those of Davies et al. (2018), Grezio et al. (2017), and Løvholt et al. (2015). BTW, what do you mean by "extant".

Page 1 – line 29: Remove "for a local area". In the end of the sentence, replace "in the area" with "in a local area".

Page 2 – line 3: Why are aggregates of buildings portfolios important in particular? Please elaborate.

Page 2- line 5: Please clarify in more detail why this is important. For instance give an example, otherwise the reader is a bit lost.

Page 2 – line 17: I'm not sure "simultaneous" is the right word. Perhaps "dependent" or spatially correlated is a better term. In any case, reformulate.

Page 3 – The simplification done by using response surfaces suppresses the uncertainty in the tsunami height (the authors uses the term wave height). This needs to be illuminated better. For instances, they could should error norms obtained using this fitting mechanism. Moreover, it needs to be clarified that tsunami heights can vary quite a bit in a local area. This property of a tsunami is concealed here, but the authors should actually quantify how large this variability is for one or more of the inundation simulations. This is important, as the authors method only operates on the fitted response function, which does not represent the full truth.

Page 3 – line 13: Is "slip ratio" the slip?

Page 3 – line 17: Statement starting with "Although tsunami numerical simulations..." is misleading. As said, fitting response functions will remove a lot of the actual variability. This needs to be explained better, otherwise it will seem that the method is better

than it actually is. . .

Page 3- line 20: As a non-expert in copula theory, this is hard to follow. Is the copula producing a unitary distribution C, mapping x to a new random variable u (with equal probability) over ui=[0,1]? Please clarified better, give more details, perhaps even a simple synthetic example. Moreover, the varible u is not even formally defined.

Page 4 – line 11: You have not introduced regions before, it is not clear what you mean. Please introduce the concept of regions. It may seem from the paper that regions refer to sources, which is quite confusing. More elaboration and clarification is needed.

Page 4 – line 15: Again essential details in the modelling is needed. The source parameters describing the focal mechnisms (slip, width, shear modulus, geometry etc) is missing. Please elaborate.

Page 4 – last paragraph: I would say that the uncertainty treatment is rather rudimentary, although some sensitivity is presented. The authors should clarify additional factors not covered by their study, such as variable (heterogeneous) slip, different possible fault configurations etc.

Page 5 – Line 9: The response surface method collapses all spatial variability into a, rather crude, single equation. In this way the uncertainty gets lost. This needs to be illuminated better. The variability from the simulations needs to be quantified.

Page 6 – line 9: "normality of the frequency distribution of the tsunami height is not secured" → "distribution of the tsunami heights do not necessarily follow a normal distribution".

Page 7 – line 18: What is [0,1] space. Be more specific. Moreover, define and introduce the AIC and BIC methods.

Page 7 – lines 24-29: Elaborate on how the different sampling technical (both with and without copulas) are carried out. For instance, you do not explain how the uncorrelated sampling is carried out.

Page 9 – line 7: I suggest that the authors explain in more detail how their findings can be used, for instance in PTHA and tsunami risk assessment. Possible use might be of value beyond the present study, but is a little bit concealed.

References

Davies, G., Griffin, J., Løvholt, F., Glimsdal, S., Harbitz, C., Thio, H. K., ... & Baptista, M. A. (2018). A global probabilistic tsunami hazard assessment from earthquake sources. Geological Society, London, Special Publications, 456(1), 219-244.

Grezio, A., Babeyko, A., Baptista, M. A., et al. (2017). Probabilistic tsunami hazard analysis: multiple sources and global applications. Reviews of Geophysics, 55(4), 1158-1198.

Løvholt, F., Griffin, J., & Salgado-Gálvez, M. (2015). Tsunami hazard and risk assessment at a global scale. Encyclopedia of complexity and systems science, 1-34.
* * *

---

## Referee Comment (RC2) · Anonymous Referee #2 · 9 Aug 2019

General comments

The paper by Fukutani et al. deals with simultaneous assessment of tsunami risk to two buildings located at a distance from each other, using spatial correlation of tsunami wave heights. The study shows that the copula modeling is useful in evaluating the tsunami risk for a portfolio of buildings. My comments are listed below.

The paper presents a good overview of specific methods used in the study, and a complete review of related literature. I cannot comment on the accuracy of the copula modeling since this is outside of my area of expertise, but the tsunami modeling methodology is solid. Results are presented in a clear way. The concept of the re-

[Figure]

sponse surface makes the probabilistic tsunami hazard assessment more efficient.

My major concern with this study is the assumption of the uniform slip on the rupture. This is never the case in a real earthquake, and it was shown in many tsunami studies that tsunami wave heights and runup values in the near field are highly sensitive to the slip distribution in the rupture area. Both towns, Oiso and Miura, are located in the near field with respect to the simulated tsunami sources (the ten Regions), and in some cases even within the rupture area of the earthquake. In my experience, the sensitivity of tsunami heights and runup values to the slip distribution is higher than that to the slip amount and depth of the fault (given that the fault depth was varied by small amounts). If the goal of the paper was to demonstrate only the proof of concept of using response surface and copulas, this needs to be stated clearly in the abstract.

Specific comments 1. It is not clear from the abstract that the considered buildings from the same portfolio are located far away from each other. It would be nice to define "portfolio of buildings" for readers who are not familiar with the civil engineering terminology. 2. A figure that shows the geographical region described in the study, including the Sagami trough, should be included. This figure can be referenced at the beginning of Section 3. 3. It is not clear why each earthquake source needs to be represented by thousands of subfaults if the slip on the rupture is uniform. 4. Technical corrections - Page 1, line 20: this sentence is not grammatically correct. - Page 2, line 27: refer to Figure 2 for locations. - Page 5, line 9: needs to be "affect" - Page 5, line 10: reference the new figure that shows the study area - Page 6, line 33: it is probably "all possible uncertainties" - Page 9, line 6: it is probably "agencies"

---

## Author Comment (AC1) · 16 Sep 2019

Dear Anonymous Referee #1,

We have considered carefully the peer-reviewed comments from you and revised our manuscript. Authors' one-on-one comments are as follows. Also, we have attached the revised manuscript as a supplement material.

We declare that this work is original and has not been published elsewhere nor is it currently under consideration for publication elsewhere. Please address all correspondence concerning this manuscript to me. Thank you for your consideration of this

manuscript.

Yo Fukutani

Authors comments to the Anonymous Referee #1

——————————————————————-

Page 1-line 24: It would preferable if key concepts of probabilistic tsunami hazard assessment (PTHA) and probabilistic tsunami risk assessment are introduced here. The authors are simply referring to them without explaining what they are. Some more details on PTHA and PTRA would be preferable.

——————————————————————-

Thank you for pointing this out. We have included additional details and references on PTHA and PTRA from Page 1 - line 25 to Page 2 - line 4 in the revised manuscript as follows:

Among them, a variety of probabilistic tsunami hazard assessment (PTHA) and probabilistic tsunami risk assessment (PTRA) methods for tsunami disasters were rapidly developed since the 2000s (e.g., Geist and Parsons, 2006; Annaka et al., 2007; González et al., 2009; Thio et al., 2010; Løvholt et al., 2012; Goda et al., 2014; Fukutani et al., 2015; Løvholt et al., 2015; Park and Cox, 2016; De Risi and Goda, 2017; Grezio et al., 2017; Davies et al., 2018). The main purpose of a PTHA is to assess the likelihood of a given measure of tsunami hazard metrics (e.g. maximum tsunami wave height) being exceeded at a particular location within a given time period. The most basic outcome of such an analysis is typically expressed as a hazard curve, which shows the exceedance level of the hazard metric with the probability. This is often expressed as a rate of exceedance per year. A PTHA can be expanded to a PTRA by combining hazard assessment with loss evaluation of a target. Several studies have proposed a method of PTRA for an individual site in a local area. Detailed risk assessment is undoubtedly important in terms of grasping the risk of exposing assets located in a local
area.
* * *
Page 1 – line 25: This study is very much on hazard as well as the risk. I suggest including hazard also in the title. Replace "the probabilistic risk. . ." with "a variety of probabilistic hazard and risk. . ."
* * *
Thank you for the advice. Based on the advice, we have changed the title of our article slightly to "Tsunami hazard and risk assessment for multiple buildings by considering spatial correlation of wave height using copulas". We also have replaced "the probabilistic risk. . ." with "a variety of probabilistic hazard and risk. . ." in Page 1 - line 25.
* * *
Page 1 – line 28: Relevant overview that preferably should be added to these references would be those of Davies et al. (2018), Grezio et al. (2017), and Løvholt et al. (2015). BTW, what do you mean by "extant".
* * *
Thank you for the comment. We have included these references in the Introduction, and we have deleted "extant" that you pointed out.
* * *
Page 1 – line 29: Remove "for a local area". In the end of the sentence, replace "in the area" with "in a local area".
* * *
Thank you for the advice. We have replaced "in the area" with "in a local area".
* * *
Page 2 – line 3: Why are aggregates of buildings portfolios important in particular? Please elaborate.

———————————————————-

Thank you for pointing this out. We have added an explanation for the importance of evaluating the detailed risks posed by aggregates of building portfolios from Page 2 - line 5 to line 10 in the revised manuscript as follows:

However, probabilistic risk evaluation methods are also utilized in cases to evaluate risks for multiple buildings. With respect to businesses that own a building portfolio, including factories and offices over a wide area, it is extremely important in risk-based management decisions to evaluate the detailed risks posed by the building portfolio. A portfolio means a collection of assets held by an institution or a private individual. By quantitatively assessing the risks posed by the building portfolio, for example, it is possible to identify assets held that have a large impact on the overall risk, and to compare the amount of risk held over time, which leads to support for decision-makers.

———————————————————-

Page 2- line 5: Please clarify in more detail why this is important. For instance give an example, otherwise the reader is a bit lost.

———————————————————-

Thank you for pointing this out. We have added some examples from Page 2 - line 13 to line 20 in the revised manuscript to help readers understand, as follows:

For example, let us consider assessing the risk of two buildings located at two sites. When the positive correlation of hazards between two sites is strong, the hazard at one site tends to be large if the hazard at another site is large. In this case, the hazards at the two target sites both increase, and as a result, the aggregate risk for the two buildings increases. Conversely, when the positive correlation of hazards is small, the hazard at one site is not necessarily large, even if the hazard at another site is large.

In this case, the aggregate risk of the two buildings is smaller than when the positive correlation of hazards is strong. Therefore, analyses that do not consider the spatial correlation of hazards involves the risk of underestimating the risk over a wide area. It is clear that the difference of aggregate risk between two cases becomes more prominent as the number of target sites increases.

——————————————————————-

Page 2 – line 17: I'm not sure "simultaneous" is the right word. Perhaps "dependent" or spatially correlated is a better term. In any case, reformulate.

——————————————————————-

Thank you for pointing this out. We have deleted "simultaneous" throughout the manuscript and used "joint" instead, which is commonly used in the statistics field.

——————————————————————-

Page 3 – The simplification done by using response surfaces suppresses the uncertainty in the tsunami height (the authors uses the term wave height). This needs to be illuminated better. For instances, they could should error norms obtained using this fitting mechanism. Moreover, it needs to be clarified that tsunami heights can vary quite a bit in a local area. This property of a tsunami is concealed here, but the authors should actually quantify how large this variability is for one or more of the inundation simulations. This is important, as the authors method only operates on the fitted response function, which does not represent the full truth.

——————————————————————-

Thank you for pointing this out. We have added details on the uncertainty of tsunami hazard assessment and additional references from Page 3 - line 23 to Page 4 - line 15 in the revised manuscript as follows:

Tsunami hazard assessment has many uncertainties in each process of tsunami generation, propagation, and run-up. Even considering only the earthquake source parameters that are the basis for calculating the initial displaced water level of the tsunami, there are fault length, fault width, fault depth, slip amount, rake, strike, and dip. The temporal and spatial changes of all these parameters more or less affect the tsunami hazard assessment. Numerous studies on the effect of earthquake source parameters on the initial displaced water level of tsunamis have been conducted (e.g., Hwang and Divoky 1970; Ward 1982; Ng et al. 1991; Pelayo and Wiens 1992; Whitmore 1993; Geist and Yoshioka 1996; Geist 1999; Song et al. 2005). These studies reported that fault slip was an important factor governing tsunami intensity. In addition, the Sagami Trough, which is the target earthquake of this study, has a complex crustal structure in the area where the Pacific Plate, the Philippine Sea Plate, and the North American Plate meet. Therefore, the depth where the Sagami Trough earthquake occurs is considered uncertain. Therefore, in this study, we decided to consider only the tsunami hazard uncertainty caused by the changes of slip amount and fault depth as an example. The heterogeneity of fault slip is an equally important factor, but we did not consider non-uniform slip distribution for purposes of simplicity. It is an important issue in the future to evaluate the heterogeneity of fault slip by response surface methodology. This is true for both slip heterogeneity and other fault parameters. For the above reasons, we model maximum tsunami wave height considering tsunami wave uncertainty with Eq. (2) after conducting tsunami numerical simulation with a nonlinear long wave equation. This formula is following the tsunami hazard evaluation method proposed by Kotani et al. (2016) that applied a reliability analysis framework using the response surface method proposed in Honjo (2011). The expression is as follows:

$h(S,D) = aS + bD + cSD + dS^2 + e$

where h (S, D) denotes the tsunami wave height, S denotes the slip, D denotes the fault depth, and a, b, c, d, and e denote the undetermined coefficients. It should be noted that an error term is not included in Eq. (2). An example of the error term is to consider an error due to modeling. For example, Kotani et al. (2016) quantified

the modeling error as the difference between the observed tsunami height and the numerically simulated tsunami height. The modeling error of the numerical analysis was also considered as one of the tsunami hazard uncertainties. However, the main purpose of this study is to propose a tsunami damage assessment method for multiple buildings using copula considering wave height correlation. Therefore, the modeling error is also ignored for simplification in this study.

Also, we have added results of the tsunami inundation simulations in Fig. 5 with explanations for the figure from Page 7 - line 14 to line 19 in the revised manuscript as follows:

As an example, Fig. 5 shows the numerical simulation results of 9 cases around Oiso and Miura in which the Mw of the source 8 is changed to $\pm$ 0.1, the fault depth is changed to + 2.0 km, and - 1.0 km. As shown in the figure, the distributions of the maximum tsunami wave height vary locally by changing the slip amount and the fault depth, and the effect of the slip amount on the maximum tsunami wave height is more dominant than the fault depth. In addition, while there is a clear positive correlation between the maximum tsunami wave height and slip amount of the earthquake, there is no clear correlation between the maximum tsunami wave height and the fault depth.

———————————————————-

Page 3 – line 13: Is "slip ratio" the slip?

———————————————————-

Thank you for pointing this out. We have modified as you noted.

———————————————————-

Page 3 – line 17: Statement starting with "Although tsunami numerical simulations. . ." is misleading. As said, fitting response functions will remove a lot of the actual variability. This needs to be explained better, otherwise it will seem that the method is better than it actually is. . .

————————————————————-

Thank you for the advice. We have added detailed explanations about the variability of tsunami hazard assessment in the second chapter to avoid appearing to mislead.

————————————————————-

Page 3- line 20: As a non-expert in copula theory, this is hard to follow. Is the copula producing a unitary distribution C, mapping x to a new random variable u (with equal probability) over ui=[0,1]? Please clarified better, give more details, perhaps even a simple synthetic example. Moreover, the varible u is not even formally defined.

————————————————————-

Thank you for pointing this out. We have added a simple synthetic example and explanation of copulas in Fig. 2 and from Page 4 - line 23 to line 26 in the revised manuscript as follows:

There exists a n-dimensional copula C such that for all x in the domain of F, the following expression holds (Sklar, 1959):

$$H(x\_1, \ldots, x\_n )=C\{F\_1 (x\_1 ),\ldots,F\_n (x\_n )\}=C(u\_1,\ldots,u\_n )$$

where $u\_i=F\_i (x\_i )âĹĹ[0,1],i=1,\ldots,n$. Figure 3 shows a simple synthetic example of a copula in a bivariate case. Fig. 3 (a) is a joint distribution function, Figs. 3 (b) and (c) are distribution functions of each variable (marginal distributions) and Fig. 3 (c) is a copula distributed over [0, 1].

————————————————————-

Page 4 – line 11: You have not introduced regions before, it is not clear what you mean. Please introduce the concept of regions. It may seem from the paper that regions refer to sources, which is quite confusing. More elaboration and clarification is needed.

————————————————————-

Thank you for pointing this out. We have decided to use the term "sources" instead of the term "regions" throughout the manuscript to clarify and avoid confusion. Please check the manuscript.

————————————————————-

Page 5 – line 15: Again essential details in the modelling is needed. The source parameters describing the focal mechanisms (slip, width, shear modulus, geometry etc) is missing. Please elaborate.

————————————————————-

Thank you for pointing this out. We have added detailed explanations of the source parameters used in this study from Page 6 - line 19 to line 27 in the revised manuscript as follows:

Each small fault corresponded to a 2.5 km square, and the slip amount of the fault was set to a uniform value based on the moment magnitude (Mw) of each earthquake by using the following scaling laws of earthquakes according to Kanamori (1977):

Mo=$\mu$SA

Mw=(log10 Mo-9.1)/1.5

where Mo denotes moment magnitude (Nm), $\mu$ denotes shear modulus (Pa), S denotes slip amount (m) and A denotes earthquake source area (m2). $\mu$ was set to 3.4 $\times$ 1010 (Pa). In this study, we did not consider non-uniform slip distribution for purposes of simplicity. We set other fault parameters (i.e., fault depth, dip, rake, and strike) to the sources based on information published by the Cabinet Office (2013) in Japan, which were created from the crustal structure of data of the plates.

————————————————————-

Page 5 – last paragraph: I would say that the uncertainty treatment is rather rudimentary, although some sensitivity is presented. The authors should clarify additional factors not covered by their study, such as variable (heterogeneous) slip, different possible fault configurations etc. Page 5 – Line 9: The response surface method collapses all spatial variability into a, rather crude, single equation. In this way the uncertainty gets lost. This needs to be illuminated better. The variability from the simulations needs to be quantified.

—————————————————————-

Thank you for pointing this out. We have described the possible uncertainty of tsunami hazard assessment in the second chapter. In addition, we have added the following sentence in the last paragraph that you pointed out. As detailed in the second chapter, this study focused on the slip amount and the fault depth among many uncertain factors.

—————————————————————-

Page 7 – line 9: "normality of the frequency distribution of the tsunami height is not secured" "distribution of the tsunami heights do not necessarily follow a normal distribution".

—————————————————————-

Thank you for the comment. We have modified as you mentioned.

—————————————————————-

Page 7 – line 18: What is [0,1] space. Be more specific. Moreover, define and introduce the AIC and BIC methods.

—————————————————————-

Thank you for pointing this out. We have changed the description to "over [0, 1]" throughout the manuscript. We also have added explanations of [0, 1] and copulas in Fig. 2 and from Page 4 - line 26 to line 29 in the revised manuscript.

———————————————————-

Page 7 – lines 24-29: Elaborate on how the different sampling technical (both with and without copulas) are carried out. For instance, you do not explain how the uncorrelated sampling is carried out.

———————————————————-

Thank you for pointing this out. We have added an explanation of how the uncorrelated sampling is carried out on Page 9 - line 13 to line 15 in the revised manuscript as follows:

To compare with this result, Fig. 11 (b) shows the results without considering the wave height correlation. We independently generated the tsunami wave height by using a uniform random number and the cumulative frequency distribution of the tsunami wave height at each site without using a copula.

———————————————————-

Page 9 – line 7: I suggest that the authors explain in more detail how their findings can be used, for instance in PTHA and tsunami risk assessment. Possible use might be of value beyond the present study, but is a little bit concealed.

———————————————————-

Thank you for the advice. We have added details on how the findings in this study can be used to Page 10 - line 21 to line 31 in the revised manuscript as follows:

In addition, the response surface method used in this study significantly reduces the numerical simulation costs for probabilistic tsunami hazard assessment considering uncertainty. In this study, we only focused on the slip amount and fault depth among many tsunami hazard uncertainties, and evaluated them using the response surface method. It has been reported that the heterogeneity of the slip distribution of the fault has a great influence on tsunami intensity. It is a future issue to evaluate these

effects with a response surface method. The evaluation result was shown for only two buildings, but when an entity evaluates the risk of assets it owns it is assumed that there will be more target sites. It is clear that as the number of target assets increases, the percentile value and maximum value of aggregate damage of assets becomes more prominent. Risk assessment that does not consider the spatial correlation of wave heights will lead to underestimation of the risks held. The basic method shown in this study can be applied even when the number of target assets increases. It is also important to avoid underestimating the assessed risk by considering the wave height correlation using a copula.

Please also note the supplement to this comment:
https://www.nat-hazards-earth-syst-sci-discuss.net/nhess-2019-139/nhess-2019-139-AC1-supplement.pdf

**Supplement:**

[revised manuscript text omitted]

---

## Author Comment (AC2) · 16 Sep 2019

Dear Anonymous Referee #2,

We have considered carefully the peer-reviewed comments from you and revised our manuscript. Authors' one-on-one comments are as follows. Also, we have attached the revised manuscript as a supplement material.

We declare that this work is original and has not been published elsewhere nor is it currently under consideration for publication elsewhere. Please address all correspondence concerning this manuscript to me. Thank you for your consideration of this.

Yo Fukutani

Authors comments to the Anonymous Referee #2

————————————————————-

My major concern with this study is the assumption of the uniform slip on the rupture. This is never the case in a real earthquake, and it was shown in many tsunami studies that tsunami wave heights and runup values in the near field are highly sensitive to the slip distribution in the rupture area. Both towns, Oiso and Miura, are located in the near field with respect to the simulated tsunami sources (the ten Regions), and in some cases even within the rupture area of the earthquake. In my experience, the sensitivity of tsunami heights and runup values to the slip distribution is higher than that to the slip amount and depth of the fault (given that the fault depth was varied by small amounts). If the goal of the paper was to demonstrate only the proof of concept of using response surface and copulas, this needs to be stated clearly in the abstract.

————————————————————-

Thank you for pointing this out. The purpose of this study is not to identify parameters that affect tsunami hazards, but to demonstrate a method for tsunami risk assessment using response surface and copulas. This has been clarified in the second sentence of the abstract. There are many parameters that affect tsunami hazards such as tsunami wave height and runup height. There are also many tsunami studies that show inhomogeneous slip has a great impact on tsunami hazards, but this is not the focal point of this study. We have added more details on the uncertainty of tsunami hazard assessment and references from Page 3 - line 23 to Page 4 - line 5 in the revised manuscript as follows:

Tsunami hazard assessment has many uncertainties in each process of tsunami generation, propagation, and run-up. Even considering only the earthquake source parameters that are the basis for calculating the initial displaced water level of the tsunami,

there are fault length, fault width, fault depth, slip amount, rake, strike, and dip. The temporal and spatial changes of all these parameters more or less affect the tsunami hazard assessment. Numerous studies on the effect of earthquake source parameters on the initial displaced water level of tsunamis have been conducted (e.g., Hwang and Divoky 1970; Ward 1982; Ng et al. 1991; Pelayo and Wiens 1992; Whitmore 1993; Geist and Yoshioka 1996; Geist 1999; Song et al. 2005). These studies reported that fault slip was an important factor governing tsunami intensity. In addition, the Sagami Trough, which is the target earthquake of this study, has a complex crustal structure in the area where the Pacific Plate, the Philippine Sea Plate, and the North American Plate meet. Therefore, the depth where the Sagami Trough earthquake occurs is considered uncertain. Therefore, in this study, we decided to consider only the tsunami hazard uncertainty caused by the changes of slip amount and fault depth as an example. The heterogeneity of fault slip is an equally important factor, but we did not consider non-uniform slip distribution for purposes of simplicity. It is an important issue in the future to evaluate the heterogeneity of fault slip by response surface methodology. This is true for both slip heterogeneity and other fault parameters. For the above reasons, we model maximum tsunami wave height considering tsunami wave uncertainty with Eq. (2) after conducting tsunami numerical simulation with a nonlinear long wave equation.

————————————————————-

Specific comments 1. It is not clear from the abstract that the considered buildings from the same portfolio are located far away from each other. It would be nice to define "portfolio of buildings" for readers who are not familiar with the civil engineering terminology.

————————————————————-

Thank you for pointing this out. We have deleted the word portfolio in the abstract in consideration of readers who are unfamiliar with the term. We have added the sentence

"it is noted that portfolio means a collection of assets held by an institution or a private individual" to the Introduction. Also, we have clearly indicated in the abstract that we evaluated buildings that are far away from each other.

————————————————————-

2. A figure that shows the geographical region described in the study, including the Sagami trough, should be included. This figure can be referenced at the beginning of Section 3.

————————————————————-

Thank you for pointing this out. We have included a new figure in Fig. 2, which includes the Sagami Trough earthquake and other major subduction earthquakes around Japan.

————————————————————-

3. It is not clear why each earthquake source needs to be represented by thousands of subfaults if the slip on the rupture is uniform.

————————————————————-

Thank you for pointing this out. In tsunami numerical simulation we commonly assume a rectangular earthquake fault. Therefore, when considering an earthquake occurrence area with a complicated shape such as the Sagami Trough earthquake, it is necessary to generate the earthquake fault for tsunami numerical simulation by aggregating thousands of rectangular subfaults, even if the slip on the rupture is uniform.

————————————————————-

4. Technical corrections - Page 1, line 20: this sentence is not grammatically correct.

————————————————————-

Thank you for pointing this out. We have modified the sentence.

————————————————————-

- Page 2, line 27: refer to Figure 2 for locations.

_________________________________-

Thank you for pointing this out. We have added the reference to Fig. 2.

_________________________________-

- Page 5, line 9: needs to be "affect"

_________________________________-

Thank you for pointing this out. We have modified it.

_________________________________-

- Page 5, line 10: reference the new figure that shows the study area

_________________________________-

Thank you for pointing this out. We have referenced the new figure in Fig.2 that clearly shows the study area.

_________________________________-

- Page 6, line 33: it is probably "all possible uncertainties"

_________________________________-

Thank you for pointing this out. We have modified it.

_________________________________-

- Page 9, line 6: it is probably "agencies"

_________________________________-

Thank you for pointing this out. We have modified it.

Please also note the supplement to this comment:
https://www.nat-hazards-earth-syst-sci-discuss.net/nhess-2019-139/nhess-2019-139-AC2-supplement.pdf

———————————————————

---

## Author Response (AR2)

Authors comments to the Anonymous Referee #1
We have considered carefully the peer-reviewed comments from you and revised our manuscript. Authors' one-on-one comments are as follows. Also, we have attached the revised manuscript.

Please address all correspondence concerning this manuscript to me. Thank you for your consideration of this manuscript.
* * *
Yo Fukutani

  Kanto Gakuin Univerity, Associate Professor

  Mutsuura Higashi 1-50-1, Kanazawa, Yokohama, Kanagawa 236-8501, Japan

Major concern:

On the last part of page 3 and first part of page 4, the authors have included a new text concerning heterogeneous slip and other complicating factors that are not treated in their analysis. This addition is highly welcomed. However, the authors missed my most important point, which concerns the spatial inundation height and flow depth variability that would emerge from a real tsunami (e.g in from field observations), or estimated through numerical inundation simulations. As said in the previous review, these variations are suppressed in the present analysis, and this issue is still not fixed. In essence, it would not help to use heterogeneous slip if the inundation field is smoothed anyway by a response surface. So a remark on this simplification is still needed. It needs to be pointed out that there will be a spatial variability not incorporated by the response surface, and that this is a limitation of the analysis. The authors touches the subject, but this was not explicitly mentioned, and you feel that the implications of this concern was not fully understood. If the authors wish, they can consult for instance Glimsdal et al. (2019), which analysed the stochastic variability of onshore flow for various numerical simulations.

Thank you for pointing this out.

We have correctly understood what you pointed out. Your indication is correct and we also consider that it is important to evaluate the spatial inundation height and flow depth variability. However, such analysis is outside the scope of this study. We have included these explanations from Page 3 - line 23 to line 25 in the revised manuscript as follows:

It should be noted here that a response surface is generated for a certain point. Therefore, it is necessary to generate a large number of response surfaces with spatial meshes in order to evaluate the spatial inundation height and flow depth variability, but such analysis is outside the scope of this study.

Minor comments

Page 2 – line 17: Thanks for providing this extra explanations. However, the sentence starting with "In this case···" is still not fully clear.

Thank you for pointing this out.

We have been changed the sentence as follows to make it clearer:

In this case, the hazards at the two target sites is smaller than when the positive correlation of hazards is strong, and as a result, the aggregate risk of the two buildings is smaller in case that the vulnerability of the two sites is equal.

Page 3 – line 28: Add reference to Geist (2002)

Thank you for pointing this out.

We have included the reference.

Page 6 – The table provide only average slip values but not average areas. Please include these as well.

Thank you for pointing this out.

We have included the areas of each source.

Page 7 – line 25: Please spell out what quantity your response surface represents.

Thank you for pointing this out. We have spelled out it as follows:

In this section, we construct response surfaces, which indicate maximum wave height at target sites.

Page 8 – line 5: Remove "the" before source 8

Thank you for pointing this out. However, there is no such word in the line you pointed out. Instead, we have removed "the" before source 8 in Page 9－line6.

---

## Author Response (AR3)

Authors comments to the NHESS editor

We have considered carefully the comments from you and revised our manuscript. Authors' one-on-one comments are as follows. Also, we have attached the revised manuscript. Thank you so much for your consideration of this manuscript.
* * *
Yo Fukutani

  Kanto Gakuin Univerity, Associate Professor

  Mutsuura Higashi 1-50-1, Kanazawa, Yokohama, Kanagawa 236-8501, Japan
* * *
(1) The sentence on page 2, line 17 still remains unclear "In this case, ..." Please, re-phrase it.

Thank you for pointing this out.

We have re-phrased it as follows:

In this case, the hazards at the two target sites both increase, and as a result, the aggregate risk for the two buildings considering the hazard correlation increases. Conversely, when the positive correlation of hazards is small, the hazard at one site is not necessarily large, even if the hazard at another site is large. In this case compared to the former case, the hazards at the two target sites are smaller, and as a result, the aggregate risk for the two buildings is smaller if we assume that the vulnerability of the two buildings is equal.
* * *
(2) In-text citations. In general, you do a good job at citing appropriate in-text citations to support your facts, but please go through the paper from beginning to end (the text and the figure captions and table headers) to make sure that EVERY sentence it is clear where the facts/information are coming from. For example, Line 5, Page 2.

Thank you for pointing this out.

We have added some citations in Line5-7, Page2 and Line25, Page2.
* * *
(3) For all figures, please ensure that they have some indication of scale either on the figure or in the figure caption. For example, in Figure 2a, 4, 5.

Thank you for pointing this out.

We have added the indication of scales in Figure 2a, 4, 5.
* * *
(4) Size of text in figures. Ensure that the minimum font size is legible in your figures. For example Figure 4b & 6 legend numbers--will these be legible once the figure is printed (would be an easy task now to enlarge the legend numbers). Figures 7, 12 the axis labels and numbers may be a bit small. The same for the Table 3 and the colorbar in Figure 5.

Thank you for pointing this out.

The minimum font size of Figure 4b & 6 could be legible.

It is difficult to modify the axis labels and numbers of Fig.7 and 12. Instead, we have the original figures with meta files. If you need, please inform us. The same for the Table 3.

We have modified the size of colorbar in Fig.5.